# Bound star clusters observed in a lensed galaxy 460 Myr after the Big Bang

Angela Adamo[1 ✉], Larry D. Bradley[2,33], Eros Vanzella[3,33], Adélaïde Claeyssens[1], Brian Welch[4,5], Jose M. Diego[6], Guillaume Mahler[7,8,9], Masamune Oguri[10,11], Keren Sharon[12], Abdurro'uf[2,13], Tiger Yu-Yang Hsiao[2,13], Xinfeng Xu[14,15], Matteo Messa[3], Augusto E. Lassen[1,16], Erik Zackrisson[17,18], Gabriel Brammer[19,20], Dan Coe[2,13,21], Vasily Kokorev[22], Massimo Ricotti[4], Adi Zitrin[23], Seiji Fujimoto[24], Akio K. Inoue[25,26], Tom Resseguier[13], Jane R. Rigby[5], Yolanda Jiménez-Teja[27,28], Rogier A. Windhorst[29], Takuya Hashimoto[30,31] & Yoichi Tamura[32]

The Cosmic Gems arc is among the brightest and highly magnified galaxies observed at redshift $z \approx 10.2$ (ref. 1). However, it is an intrinsically ultraviolet faint galaxy, in the range of those now thought to drive the reionization of the Universe[2-4]. Hitherto the smallest features resolved in a galaxy at a comparable redshift are between a few hundreds and a few tens of parsecs (pc)[5,6]. Here we report JWST observations of the Cosmic Gems. The light of the galaxy is resolved into five star clusters located in a region smaller than 70 pc. They exhibit minimal dust attenuation and low metallicity, ages younger than 50 Myr and intrinsic masses of about $10^6 M_\odot$. Their lensing-corrected sizes are approximately 1 pc, resulting in stellar surface densities near $10^5 M_\odot \, pc^{-2}$, three orders of magnitude higher than typical young star clusters in the local Universe[7]. Despite the uncertainties inherent to the lensing model, they are consistent with being gravitationally bound stellar systems, that is, proto-globular clusters. We conclude that star cluster formation and feedback likely contributed to shaping the properties of galaxies during the epoch of reionization.

The Cosmic Gems arc (SPT0615-JD1) was initially discovered in Hubble Space Telescope (HST) images obtained by the Reionization Lensing Cluster Survey (RELICS) of the lensing galaxy cluster SPT-CLJ0615−5746 at $z = 0.972$ and reported as a redshift $z = 10$ candidate[8].

A recent James Webb Space Telescope Near Infrared Camera (JWST/NIRCam) imaging campaign of this field has observed the Cosmic Gems arc with eight bands covering the 0.8–5.0 μm range (Methods). Spectral energy distribution (SED) fitting to the James Webb Space Telescope (JWST) photometry indicates that the Cosmic Gems galaxy has a fairly young stellar population with a mass-weighted age of less than 79 Myr and a lensing-corrected stellar mass in the range of $2.4–5.6 \times 10^7 M_\odot$, with low dust extinction ($A_V < 0.15$ mag) and metallicity (<1% $Z_\odot$) (ref. 1).

The far-ultraviolet (FUV)-to-optical rest frame of Cosmic Gems arc shows bright clumpy structures and extended faint emission over a 5″-long arc (Fig. 1). The symmetry between the south–east (hereafter Img.1) and the north–west (Img.2) part of the arc uncovers two lensed mirror images of the galaxy, implying that the Cosmic Gems arc is observed at very high magnification on the lensing critical curve. Four independent magnification models have been created to account for the galaxy appearance. All the models successfully reproduce the $z = 10.2$ critical line crossing the Cosmic Gems arc (Methods).

Five star cluster candidates are uniquely identified in Img.1. In Img.2, three sources are distinguishable in the F150W filter (Fig. 1), along with a fourth fainter source (E.2). The appearance of Img.2 is probably perturbed by further lensing effects because of the northern galaxy at $z \approx 2.6$ (visible in the top right corner) and possibly by an undetected small scale perturber closer to the arc[1]. Source D.2 is possibly blended with C.2 and is therefore identified for the remaining of the analysis

[1]Astronomy Department, Stockholm University and Oskar Klein Centre, Stockholm, Sweden. [2]Space Telescope Science Institute (STScI), Baltimore, MD, USA. [3]Osservatorio di Astrofisica e Scienza dello Spazio di Bologna, INAF, Bologna, Italy. [4]Department of Astronomy, University of Maryland, College Park, MD, USA. [5]Astrophysics Science Division, Code 660, NASA Goddard Space Flight Center, Greenbelt, MD, USA. [6]Instituto de Física de Cantabria, CSIC-UC, Santander, Spain. [7]STAR Institute, Liège, Belgium. [8]Centre for Extragalactic Astronomy, Durham University, Durham, UK. [9]Institute for Computational Cosmology, Durham University, Durham, UK. [10]Center for Frontier Science, Chiba University, Chiba, Japan. [11]Department of Physics, Graduate School of Science, Chiba University, Chiba, Japan. [12]Department of Astronomy, University of Michigan, Ann Arbor, MI, USA. [13]Center for Astrophysical Sciences, Department of Physics and Astronomy, The Johns Hopkins University, Baltimore, MD, USA. [14]Department of Physics and Astronomy, Northwestern University, Evanston, IL, USA. [15]Center for Interdisciplinary Exploration and Research in Astrophysics (CIERA), Northwestern University, Evanston, IL, USA. [16]Instituto de Física, Departamento de Astronomia, Universe Federal do Rio Grande do Sul, Porto Alegre, Brazil. [17]Observational Astrophysics, Department of Physics and Astronomy, Uppsala University, Uppsala, Sweden. [18]Swedish Collegium for Advanced Study, Uppsala, Sweden. [19]Cosmic Dawn Center (DAWN), Copenhagen, Denmark. [20]Niels Bohr Institute, University of Copenhagen, Copenhagen, Denmark. [21]Association of Universities for Research in Astronomy (AURA) for the European Space Agency (ESA), STScI, Baltimore, MD, USA. [22]Kapteyn Astronomical Institute, University of Groningen, Groningen, The Netherlands. [23]Department of Physics, Ben-Gurion University of the Negev, Be'er-Sheva, Israel. [24]Department of Astronomy, The University of Texas at Austin, Austin, TX, USA. [25]Department of Physics, School of Advanced Science and Engineering, Faculty of Science and Engineering, Waseda University, Tokyo, Japan. [26]Waseda Research Institute for Science and Engineering, Faculty of Science and Engineering, Waseda University, Tokyo, Japan. [27]Instituto de Astrofísica de Andalucía, (CSIC), Granada, Spain. [28]Observatório Nacional, (MCTI), Rio de Janeiro, Brazil. [29]School of Earth and Space Exploration, Arizona State University, Tempe, AZ, USA. [30]Division of Physics, Faculty of Pure and Applied Sciences, University of Tsukuba, Tsukuba, Japan. [31]Tomonaga Center for the History of the Universe (TCHoU), University of Tsukuba, Tsukuba, Japan. [32]Department of Physics, Graduate School of Science, Nagoya University, Nagoya, Japan. [33]These authors contributed equally: Larry D. Bradley, Eros Vanzella. ✉e-mail: angela.adamo@astro.su.se

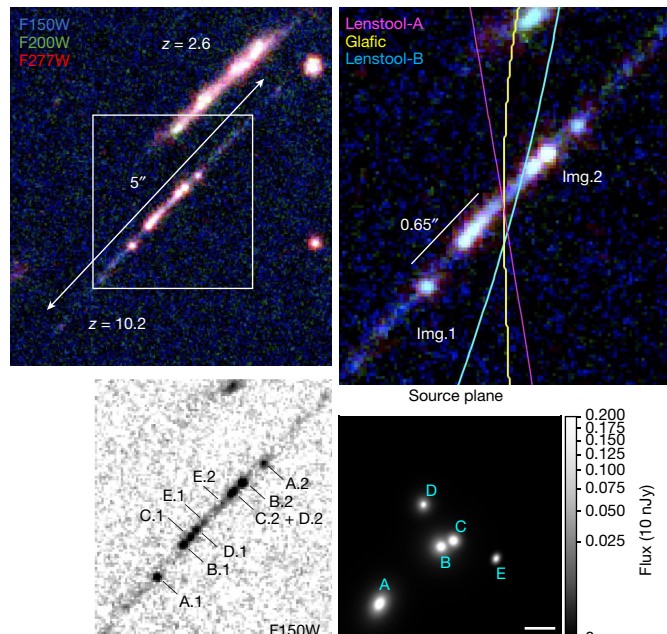

**Fig. 1 | The Cosmic Gems arc in a JWST colour composite.** The filter combination shows the rest-frame UV, blue optical wavelengths (1,200–2,800 Å). The arc is extended over 5″. A foreground galaxy at redshift $z_{phot} = 2.6$ is visible above and to the right. The field of view is rotated North up. Top right, a magnified inset of the centre of the arc in which the brightest star clusters are located, highlighted by the white square on the left image. Two mirror images are observed because of gravitational lensing. Lensing critical curves based on three models are shown bisecting the arc. Bottom left, each star cluster is labelled in a grey scale FUV rest-frame image of the galaxy. Bottom right, source plane reconstruction of the core of the galaxy in which the star clusters are located showing their relative sizes and positions. The physical distance between A and E and A and D is about 40 pc. Note that there is some uncertainty in the source positions parallel to the lensing caustic. Scale bar, 10 pc (bottom right).

as C.2 + D.2. The source E.2 is only detected at a $2\sigma$ level and not included in this analysis (Methods). The observed projected distance between the A.1 and E.1 clusters in Img.1 is about 0.65″. Using forward modelling and Lenstool-A predictions presented in the Methods, we find that their physical distance is $42^{+29}_{-5}$ pc. The star clusters are all located within this compact region (Fig. 1). The size of this region is similar to that reported for individual stellar clumps observed in the moderately lensed galaxies at redshift $z \approx 10$ (refs. 2,5), suggesting that they likely harbour star clusters within them.

The intrinsic physical properties of these five star clusters are particularly meaningful for probing proto-globular cluster formation mechanisms as well as their potential evolution. As described in the Methods, we find that A.1 is only marginally resolved, with an observed effective half-light radius $R_{eff,obs} = 0.6$ px, whereas C.1 and B.2 are consistent with being unresolved. For the latter sources, we assumed an upper limit to their radii coincident with the half-width half maximum (HWHM) of the stellar PSF in the F150W (0.025″ = 1.25 px). To derive lensing-corrected $R_{eff}$, we assumed the predicted *Lenstool-A* tangential magnifications at the location of the star clusters. The five star clusters have intrinsic $R_{eff}$ close to 1 pc (within uncertainties; Table 1). Using the other lensing models produces similar size ranges (0.3–0.9 pc for Lenstool-B and 0.3–1.2 pc for Glafic). Independent intrinsic sizes ($R_{eff,FM}$ in Table 1) have been derived by projecting the star cluster shapes from the source plane into the image plane. The latter method recovered intrinsic sizes in excellent agreement, within the uncertainties, with those measured in the image plane, strengthening the reliability of the derived values.

## Table 1 | Estimated physical properties of the Cosmic Gems arc star clusters

| ID | $R_{eff,obs}$ | $R_{eff}$ | $R_{eff,FM}$ | $M_{*,int}$ | $\Sigma_*$ | $\log(\Pi)$ |
|---|---|---|---|---|---|---|
| | (px) | (pc) | (pc) | $(10^6 M_\odot)$ | $(10^5 M_\odot\,\mathrm{pc}^{-2})$ | |
| A.1 | $0.6^{+0.4}_{-0.1}$ | $1.1^{+0.7}_{-0.2}$ | $1.1 \pm 0.1$ | $2.45^{+5.20}_{-1.56}$ | $1.92^{+1.60}_{-1.44}$ | $1.94^{+0.71}_{-0.27}$ |
| B.1 | $1.1^{+0.1}_{-0.5}$ | $1.1^{+0.1}_{-0.5}$ | $0.9 \pm 0.1$ | $2.65^{+1.09}_{-1.26}$ | $1.93^{+4.16}_{-1.11}$ | $2.11^{+0.83}_{-0.50}$ |
| C.1 | <1.25 | <1 | $0.9 \pm 0.2$ | $1.13^{+1.77}_{-0.65}$ | >1.3 | >1.90 |
| D.1 | $1.2^{+0.2}_{-1.1}$ | $0.6^{+0.1}_{-0.6}$ | $0.8 \pm 0.2$ | $1.13^{+1.23}_{-0.74}$ | $2.39^{+7.41}_{-1.98}$ | $2.17^{+0.85}_{-1.03}$ |
| E.1 | $1.5^{+0.7}_{-0.5}$ | $0.4^{+0.2}_{-0.1}$ | $0.7 \pm 0.2$ | $1.01^{+0.37}_{-0.36}$ | $6.92^{+4.90}_{-4.22}$ | $3.06^{+0.32}_{-0.59}$ |
| A.2 | $1.0^{+0.4}_{-0.3}$ | $1.7^{+0.8}_{-0.4}$ | $1.3 \pm 0.1$ | $2.89^{+1.56}_{-1.35}$ | $0.88^{+0.98}_{-0.46}$ | $1.94^{+0.52}_{-0.41}$ |
| B.2 | <1.25 | <1.4 | $1.0 \pm 0.04$ | $3.01^{+3.21}_{-1.61}$ | >5.10 | >1.8 |
| C.2+D.2 | $2.7^{+16.1}_{-2.6}$ | $1.9^{+11.4}_{-1.9}$ | $0.9 \pm 0.1$ | $4.36^{+0.98}_{-1.90}$ | $1.05^{+7.89}_{-0.89}$ | $1.99^{+1.12}_{-1.49}$ |

We report de-convolved observed half-light radii in pixels, $R_{eff,obs}$, lensing-corrected $R_{eff}$ and median stellar masses $M_{*,int}$ using magnifications produced by the reference Lenstool-A model. $M_*$ are recovered from the BAGPIPES-exp reference fit. Errors are estimated from 68% confidence level of the distributions. These quantities have been used to determine stellar surface density, $\Sigma_*$, and dynamical age $\Pi$ listed in the last columns. The evaluation of magnification uncertainties is discussed in the Methods.

The star clusters have been fitted with BAGPIPES[9] and PROSPECTOR[10]. We tested different star formation history (SFH) assumptions that simulate a single burst (inherent to the small sizes of the stellar systems analysed), different high-mass limits of the initial mass function (IMF) and models with stellar binaries (Methods). Despite the assumptions, the resulting physical properties of the clusters (ages, masses, extinction and metallicities) are in reasonable agreement. In the analysis presented here, we use the physical values derived with an SFH based on a single exponential decline with $\tau = 1$ Myr (referred to as BAGPIPES-exp, Extended Data Table 2).

The recovered ages of the star cluster candidates are between 9 Myr and 36 Myr. The age range suggests that star formation has been propagating within this compact area of the galaxy for a few tens of Myr. The measured rest-frame UV slopes of the star clusters ($\beta$ between −1.8 and −2.5, with $F_\lambda \propto \lambda^\beta$; Extended Data Table 1) are similar to those found for more evolved star clusters in the Sunburst arc at redshift 2.37 (ref. 11). Although the Cosmic Gems clusters are not extremely young, they have likely delivered large amounts of energy and momentum to their host galaxy.

The lensing-corrected stellar masses range between $1.0 \times 10^6 M_\odot$ and $2.6 \times 10^6 M_\odot$, for a total combined stellar mass of $8.3 \times 10^6 M_\odot$. The total mass of the clusters is close to 30% of the total stellar mass of the host. As the mass-weighted age of the galaxy and those of the star clusters are comparable, we can extrapolate the cluster formation efficiency (CFE)[12] to be around 30%. A caution note is necessary because the mass estimates (both for the galaxy and star clusters) are subjected to magnification values and SED fit uncertainties, making the quoted CFE uncertain. A more direct way to establish the CFE is to use the fraction of observed FUV light in star clusters with respect to the host. This quantity is not affected by the same degeneracy as the mass estimates and, thus, is a more reliable indicator of the CFE, under the assumption that the FUV light is produced by stellar populations formed during a similar timescale (as we find here). The analysed star clusters account for about 60% of the total F150W flux of the host extracted within an elliptical Kron aperture (0.51 ± 0.01 μJy, corresponding to an intrinsic FUV ABmag of −17.8 after lensing correction[1]), thus reinforcing the conclusion that star formation in star clusters is the main mode for the Cosmic Gems arc and high-redshift galaxies with similar physical properties. This observationally driven conclusion is supported by high-resolution numerical simulations[13] and analytical models[14] that find that compact star clusters with sizes of 0.5–2 pc are the dominant star formation mode in the first low-metallicity dwarf galaxies.

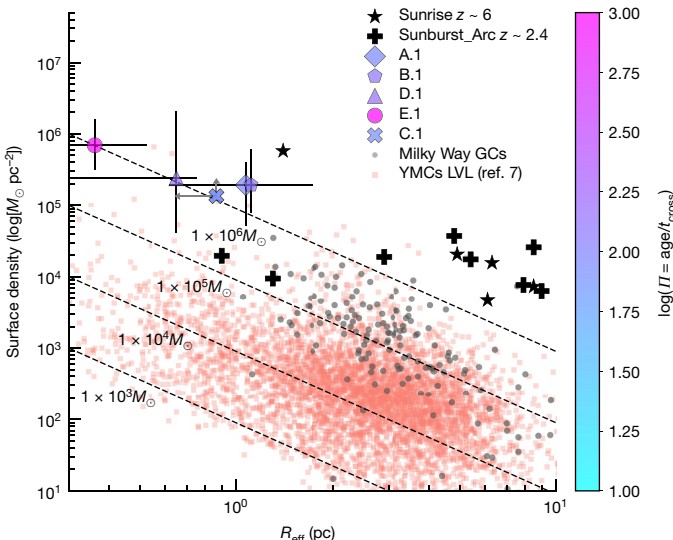

**Fig. 2 | Cluster stellar surface density versus half-light radius $R_{eff}$.** The star clusters in the $z_{phot} \approx 10.2$ Cosmic Gems arc are shown colour-coded by their dynamical age, $\log(\Pi)$. The plotted values have been derived using the reference lensing model. Predictions by other models do not change the observed trends. The error bars do not account for magnification uncertainties. However, refer to Extended Data Fig. 4, in which we show that magnification uncertainties do not affect these results. Other gravitationally bound star clusters detected in lensed galaxies at $z \approx 6$ (ref. 18) and $z = 2.38$ (ref. 19) are included, along with the $z = 0$ Milky Way globular clusters [20] and young star cluster properties of star-forming spiral galaxies in the Local Volume (distance < 16 Mpc, ref. 7). Lines of equal mass show the change in density as a function of $R_{eff}$. GCs, globular clusters; YMCs LVL, young massive clusters in the Local Volume.

The compactness of the star clusters also seems to drive the leakage of hydrogen ionizing radiation from their natal molecular cloud[15], making the star clusters observed here potential contributors to cosmic reionization. Massive star clusters similar to those observed in the Cosmic Gems arc are predicted by the feedback-free starburst model in ref. 16 and could be at the root of the super-Eddington conditions necessary to launch strong outflows in short timescales[17], both models aimed to explain the bright UV luminosity reported for $z > 9$ galaxies.

The resulting stellar surface densities of the Cosmic Gems are around $10^5 M_\odot$ pc$^{-2}$ (Table 1). Consistent physical properties have been reported in star clusters detected in the Sunrise arc at $z \approx 6$ (ref. 18) and the Sunburst arc at $z = 2.37$ (ref. 19) (Fig. 2). Using the derived ages, masses and intrinsic sizes, we also determine whether these stellar systems are gravitationally bound. According to the framework introduced in ref. 20, a star cluster is considered bound if its age is greater than the crossing time of the system (where $t_{cross} = 10\sqrt{R_{eff}^3/GM}$), or in other words, under the assumption of virial equilibrium, a cluster is gravitationally bound if $\Pi = \text{Age}/t_{cross} > 1$. The $\log(\Pi)$ values reported in Table 1 are all significantly larger than unity, indicating that we are detecting gravitationally bound star clusters in an early galaxy, 460 Myr after the Big Bang. This conclusion is valid in spite of the uncertainties inherent to physical quantity estimates as well as lensing models.

The Cosmic Gems arc clusters (Fig. 2) have substantially higher stellar densities and smaller sizes than typical young star clusters observed in the local Universe[7] as well as global clusters in the Milky Way[21]. The offset with respect to young star clusters in the local Universe is expected because the conditions under which star formation operates in reionization-era galaxies are more extreme (for example, galaxies are more compact, harbour harder ionizing radiation fields, and reach higher electron densities and temperatures[6,22]). The offset with respect to local global clusters could be explained in terms of dynamical evolution. Global clusters are hot stellar systems in which

stars continuously exchange energy and momentum. Three different internal mechanisms contribute to their dynamical evolution over a Hubble time: (1) mass loss due to stellar evolution; (2) relaxation due to $N$-body interactions; and (3) formation and dynamics of stellar black holes (SBHs)[23,24]. Mass loss due to stellar evolution drives the adiabatic expansion of global clusters under the condition of virial equilibrium. A typical mass loss of 50% will expand the initial radius of the Cosmic Gems proto-globular clusters by a factor of 2, whereas the density will decrease correspondingly by a factor of 8 (ref. 23). Relaxation time scales (shortened by the presence of SBHs[25]) will also contribute to their expansion. Finally, external tidal fields will further affect the dynamical evolution of these bound stellar systems, which appear to be bona fide proto-globular clusters.

Very dense stellar clusters ($\Sigma_* \approx 10^5 M_\odot$ pc$^{-2}$, which for $R_{eff} = 1$ pc correspond to $\rho_h \approx 10^5 M_\odot$ pc$^{-3}$), similar to those detected in the Cosmic Gems arc, are predicted to form in low metallicity and highly dense gas[26], in which radiative pressure cannot counteract the collapse, resulting in extremely high star-formation efficiencies (about 80%; ref. 27). The high stellar densities found in these proto-globular clusters imply a notable increase in stellar black hole mergers in their interiors[28,29] and therefore pave the way to intermediate-mass black hole seeds[30]. With stellar masses greater than $10^5 M_\odot$, these star clusters naturally harbour Wolf–Rayet and very massive stars[31], and because of their elevated stellar densities, satisfy the necessary condition to form supermassive stars in runaway collisions within their cores[32]. These different classes of stars are among the potential polluters that could explain the observed nitrogen enrichment in the ionized gas of high-redshift galaxies[33], possibly linked to the formation of the chemically enriched stellar populations ubiquitously found in Milky Way global clusters[34].

Cosmological simulations that focus on Milky Way disk-like assembly find that most of its global cluster population form at redshift $z < 7$ (refs. 35–37), suggesting that these star clusters forming at $z \approx 10$ might build up the global cluster populations of more massive early-type galaxies in the local Universe. It is difficult to predict whether the proto-globular clusters of the Cosmic Gems arc will survive a Hubble time. Their chances would be highly enhanced if they were ejected into their host halo during dynamical interactions[12].

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

## Methods

### Size and flux measurements

The JWST/NIRCam[38] observations of the SPT-CL J0615–5746 galaxy cluster were obtained in 2023 September (GO 4212: principal investigator Bradley) using four short-wavelength (SW) filters (F090W, F115W, F150W and F200W) and four long-wavelength (LW) filters (F277W, F356W, F410M and F444W) spanning 0.8–5.0 μm. Each filter had an exposure time of 2920.4 s. The data were reduced with the GRIZLI (v.1.9.5) reduction package[39]. They suffered from strong wisps[40] and required special background subtraction as described in ref. 1. The final data are in units of 10 nJy. The NIRCam SW images were drizzled to a pixel scale of 0.02″ per pixel and the LW images were drizzled to a pixel scale of 0.04″ per pixel. For further details on the observations and image reduction, please see ref. 1. We assume throughout the analysis a cosmology with $H_0 = 67.7$ km s$^{-1}$ Mpc$^{-1}$ and $\Omega_m = 0.31$ (ref. 41). Under these assumptions, 1 SW pixel (0.02″) corresponds to 83.7 pc at $z = 10.2$.

We derived the star cluster radii and multiband photometry by applying the method published and tested in refs. 42–44. We simultaneously fitted for the shape of the light distribution, the flux and the local background (including the galaxy diffuse light) of each of the identified clusters in the reference filter, F150W, which offers the sharpest view of the clusters. Empirical PSFs in all filters used in this work were built by selecting stars in each band. We fitted the PSF out to 0.4″ with an analytical expression, which was then convolved with varying two-dimensional (2D) Gaussians to create a grid of models. Intrinsic sizes (de-convolved by PSF) have then been derived by fitting the observed light distribution of each cluster with this grid of 2D Gaussian models in the F150W. Owing to the large shear effects, compact sources might appear resolved in the shear direction ($y_{std}$) (ref. 45). We assumed that the measured major-axis $y_{std}$ of the 2D Gaussian ellipse is the standard deviation of a 2D circular Gaussian that we translated into the observed PSF-deconvolved effective radius, $R_{eff,obs} = y_{std} \times \sqrt{2 \ln(2)}$. The derived $R_{eff,obs}$ are reported in Table 1. The de-lensed intrinsic effective half-light radii, $R_{eff}$, have been determined by dividing $R_{eff,obs}$ by the tangential magnification $\mu_{tan}$ (Extended Data Table 3).

For each star cluster, the flux in the reference filter has been determined by integrating the fitted shape and subtracting the local background. We then measured the fluxes in the other bands by convolving the derived intrinsic shape in F150W with the empirical PSF of the respective bands. Fitting this model to the other bands, we let the centre, normalization and local background (including the galaxy diffuse light) as free parameters.

The intrinsic sizes and observed fluxes in the reference filter F150W were derived using a cutout box centred on the source with a size of $11 \times 11$ px (about four times the FWHM) for A.1 and A.2. Larger box sizes did not produce noticeable differences in the output sizes and fluxes. Owing to their proximity, B.1, C.1, D.1 and E.1 have been fitted simultaneously within a box of $15 \times 15$ px. A larger box size produces consistent values within the uncertainties of measurements for B.1 and C.1, whereas E.1 gets increasingly elongated, affecting the fit of D.1. To avoid this degeneracy, we fix the box size to $15 \times 15$, which would correspond to fix the source ellipticity of the faint E.1 to 2. Similarly, B.2, C.2 and D.2 were fitted simultaneously within a box of $11 \times 11$ (changing the box size does not produce noticeable effects on the recovered parameters). We did not detect two maxima at the location of C.2 and D.2, so we allowed the fit to optimize the centre of a second hidden source. We also repeated the fit of this region by assuming only one source. Both approaches produced similar residuals. The flux extracted by assuming only one source is comparable within uncertainties to the flux extracted by fitting for C.2 and D.2. Owing to the degeneration in identifying the position of D.2, we extracted the physical properties by fitting only one source that we refer to as C.2 + D.2. Owing to the faintness of E.2 ($2\sigma$), our method did not produce meaningful constraints.

We, therefore, excluded E.2 from our analysis. In the BAGPIPES-exp fit, we find that C.1 and D.1 have similar ages and a combined total mass of about $2 \times 10^6 M_\odot$. C.2 + D.2 has a slightly older age (but in agreement within $1\sigma$) than C.1 and D.1. The total mass and size of C.2 + D.2 is a factor of two higher than their counterparts, corroborating the idea that the two star clusters are blended in Img.2.

The size uncertainties were derived by bootstrapping the fit of the source taking into account the RMS of the local background. The photometric errors include the latter uncertainties as well as the sum in quadrature of the local background variance estimated within the box in which the sources have been fitted. Aperture corrections have been extrapolated up to 0.4″ in all bands.

In Extended Data Fig. 1, we show the best model of the star clusters and the residual image in the reference filter and two more bands. The extraction of the sources does not produce significant artificial residuals above the RMS of the image.

Independent measurements of intrinsic $R_{eff}$ have been obtained following the forward modelling method in ref. 46. Briefly, this method creates a model of the galaxy in the source plane and then projects that model into the image plane. After convolving with the measured empirical PSF, the image plane model is compared with the observed data. The source plane model parameters are first optimized using a downhill simplex algorithm, then sampled using an MCMC with the Python package emcee[47].

For SPT0615-JD1, the source plane model consisted of five Sersic profiles centred on the five identified clumps A.1–E.1 (Extended Data Fig. 2). No diffuse component of the arc has been included in this analysis because of the faintness of this component with respect to the clusters. Separately, we modelled clumps A.2–C.2 on the other side of the lensing critical curve. Uncertainties in the Lenstool-A lens model resulted in slight offsets between the source plane positions of clumps on the opposite sides of the critical curve, which prevents simultaneous fitting of the two images of the arc. We found similar results for clump sizes on both sides of the critical curve, with clump radii ranging from 0.7 pc to 1.1 pc (Table 1).

### SED fitting analysis

We performed SED fitting with BAGPIPES[9] and tested the derived physical properties against different assumptions, as well as with a different software PROSPECTOR[10]. For all the runs, we fixed the redshift at $z = 10.2$, as measured by ref. 1. The standard stellar population templates were reprocessed with CLOUDY to generate nebular continuum and line information (see ref. 48 for comparisons of the two code implementations). In both codes, we assumed a Kroupa IMF, unless otherwise specified. We constrained SFHs to prescriptions that reproduce a short burst in all tests except the one in which $\tau$ is set free to vary. The short burst assumption is in agreement with the studies of stellar cluster and global cluster populations in the local Universe (refs. 12,34). The recovered median of the posterior distributions of age, mass, $A_V$, metallicity and associated 68% uncertainties are reported in Extended Data Table 2. We let the ionization parameter, $U$, to change between −2 and −3.5. We assumed a Calzetti attenuation[49] but tested also the SMC extinction. In the reference set, used to produce results reported in Fig. 2 and Table 1 and referred to as BAGPIPES-exp, we assumed an exponential decline with a very short $\tau = 1$ Myr and Calzetti attenuation. Extended Data Fig. 3 shows the observed SEDs of the five star clusters identified in Img.1 (black dots with uncertainties). When available, we include the observed SED of the corresponding clusters in Img.2 (orange stars with associated errors). The latter have been normalized by the median flux ratio in the six bands of the corresponding source in Img.1 to match the flux level while preserving the intrinsic SED shape. The best spectral and integrated photometry model obtained for the BAGPIPES-exp fit is included. The overall shapes of the observed SEDs of mirrored clusters in both images are similar within uncertainties, confirming the symmetry.

To check the consistency of the derived physical properties of the cluster, we made different assumptions. The outputs are summarized in Extended Data Table 2 in which we list the median and 68% values produced by the different fits. Changing the attenuation prescription from Calzetti to SMC produces noticeably smaller $A_v$, but all the recovered parameters are still within the 68% uncertainties associated with the recovered values. In BAGPIPES-burst, we assumed a single burst. We recovered slightly older ages and larger masses (but noticed uncertainties) that would prefer higher stellar surface densities and older dynamical ages, confirming that we were looking at dense and bound star clusters. In a third SED fitting set, BAGPIPES-BPASS, we used BPASS v.2.2.1 SED templates[50] and the fiducial BPASS IMF with the maximum stellar mass of $300M_\odot$ and a high-mass slope similar to that in ref. 51. Also, this model reproduces values that are very close to the reference value, suggesting that the clusters are compatible with being slightly older and therefore less sensitive to the presence of very massive stars and binary systems in their light (and the limitation of fitting only six broad and medium bands covering FUV-blue optical). Letting $\tau$ = free (we report mass-weighted parameters in Extended Data Table 2) produces significantly older ages, but similar masses, thus not affecting the results presented in this study.

PROSPECTOR allows us to test single stellar population SFH. In this case, we find that the age of A.1 is slightly younger (but within uncertainties) than those produced by the BAGPIPES, resulting in lower masses. This would result in slightly lower intrinsic mass $M_{*,int} = 0.39 \times 10^6 M_\odot$, $\log(\Sigma_*) = 4.5 M_\odot \, pc^{-2}$, and $\log(\Pi) = 1.2$, but leaving unchanged any of the conclusions of this study.

Finally, given that some of the knots in the Cosmic Gems arc are unresolved (C.1 and B.2) or only marginally resolved (A.1) in our current images, we have also explored scenarios in which these sources are individual, highly magnified stars. Using SED models for stars at high redshifts[52] we find that, although the slopes of the SEDs of these sources would be broadly consistent with individual stars at effective temperatures greater than about 20,000 K, these scenarios would require magnifications well in excess of what our macrolens models predict at the positions of these sources. Even the most massive and luminous stars (initial mass $560-575M_\odot$) described by the stellar evolutionary tracks of ref. 53 would require magnifications $\mu > 1,000$ to explain the observed fluxes of C.1, B.2 or A.1.

## Lens models and uncertainties on the derived star cluster physical properties

Four different lensing models have been created for the SPT-CL J0615−5746 cosmological field. The models are presented in detail in ref. 1. We include here below a short description.

Lenstool-A, here used as a reference model for the analysis presented in this Article, is based on the software LENSTOOL[54], which uses a parametric approach and MCMC sampling of the parameter space to identify the best-fit model and uncertainties. In Lenstool-A, we model the cluster lens as a combination of three main halos and cluster member galaxies, all parameterized as pseudo-isothermal mass distributions. The model uses as constraints the positions of 43 multiple images of 14 clumps, belonging to 9 unique source galaxies. The redshifts of three sources are used as constraints (the $z = 10.2$ arc, and sources at $z = 1.358$ and $z = 4.013$; ref. 55), whereas the rest of the redshifts are treated as free parameters. Three clumps on each side of the main arc were used as constraints, A, B and C, assumed to be at $z = 10.2$. The model predicts a counterimage at (right ascension (R.A.), declination (Decl.)) = (93.9490607, −57.7701814). A possible candidate of this counterimage, observed near this location (about 1.8″), was not used as a constraint. The image plane RMS of the best-fit model is 0.36″. All the observed lensed features are well reproduced by this model.

The second model, here referred to as Lenstool-B, uses the same algorithm, but with noticeably different assumptions. This model uses 43 multiple images from 11 unique sources. A secondary galaxy cluster scale halo is placed around the location of dusty galaxies nearly 50 arcsecs north of the bright centre galaxy and allowed to move within a 20″ box around this position. As with the previous model, the $z = 10.2$ arc has a predicted counterimage near the possible candidate and is only about 2″ away from the Lenstool-A model. The main differences between these models are the assumptions about the mass distribution of the lens and the addition of constraints. The image plane RMS of the best-fit model is 0.68″.

The third model used in this analysis has been created with Glafic. The Glafic[56,57] mass model is constructed with three elliptical NFW[58] halos, external shear and cluster member galaxies modelled by pseudo-Jaffe profile. The model parameters are fitted to reproduce the position of 44 multiple images generated from 15 background sources. Spectroscopic redshifts are available for 7 of the 15 sources. We include positions of A.1/A.2 and B.1/B.2 in the Cosmic Gems arc as constraints, with small positional errors of 0.04″ to accurately predict the magnifications of each star cluster image. For the other multiple images, we adopt the positional error of 0.4″. Our best-fitting model reproduces all the multiple image positions with the RMS of image positions of 0.41″.

As a consistency check, we excluded the positional constraints from the Cosmic Gems arc to construct the mass model and confirmed that the critical curve of this mass model still passes through the arc. Our Glafic best-fitting mass model also predicts a counterimage of the Cosmic Gems arc at around (R.A., Dec.) = (93.9504865, −57.7696559). We find that there is a candidate counterimage at around 2″ from the predicted position, (R.A., Dec.) = (93.9500002, −57.7702197). Both the consistency check and the presence of the candidate counterimage confirm the validity of this mass model.

A fourth model has also been produced with WSLAP+[59,60]. The WSLAP+ lens models offer an alternative to parametric models and are free of assumptions made about the distribution of dark matter. When the $z = 10.2$ arc is not included as a constraint, the WSLAP+ model predicts the critical curve passing at about 1″ from the $z = 10.2$ arc. This solution predicts a mirrored image of the arc that is not observed, reinforcing the expectation that the Cosmic Gems is a double image with the critical curve passing through the middle. When the arc is included as a constraint, the predicted critical curve passes between C.1 and D.1, just 0.3″ from the alleged symmetry point in the arc and within the uncertainties typical of WSLAP+ models. Moreover, this model predicts the position of a third counterimage consistent with the previous models. This model is currently under development with the goal of explaining the perturbation seen in Img.2 and is therefore not included in this analysis.

The photometric redshift of the candidate counter image is $z_{phot} = 10.8^{+0.6}_{-1.4}$ (95% confidence)[1], in agreement with the expectation.

The total and tangential magnifications at the position of the star clusters, $\mu_{tot} = \mu_{tang} \times \mu_{rad}$, are reported in Extended Data Table 3. For the two Lenstool-based models, we estimated uncertainties following the method presented in ref. 44 based on magnification maps produced from the Lenstool MCMC posterior distributions of the lens model. Uncertainties are omitted for the Glafic model.

In Extended Data Fig. 4, we show the impact that magnification predictions have in the recovered physical properties (intrinsic half-light radius and mass, black and blue solid lines) and derived quantities (dynamical age and stellar surface density in magenta and orange solid lines). We use logarithmic scales so that all quantities can be included. The coloured bands show the level of uncertainties recovered from the analysis. We also include the upper limits on the $R_{eff}$ (assuming that the source is unresolved and has a size smaller than the stellar PSF) and what type of lower limits it will translate for the physical quantities that depend on the size estimates as dashed lines (notice that these are the reference quantities for C.1 that is unresolved). The magnifications (total in the bottom image and tangential in the top image) are reported on the x-axis. As we move to lower magnifications, the derived masses

and radii become larger, consequently predicting lower stellar surface densities and dynamical ages. However, even in the unlikely case that the magnifications are wrong by one order of magnitude, the stellar surface density will remain above $10^4 M_\odot$ pc$^{-2}$ and dynamical ages will still be significantly larger than 1 ($\log(\Pi) > 0$), leaving the main conclusion of this analysis unchanged: we are detecting bound proto-globular clusters within the first 500 Myr of our Universe.

## Data availability

The data were acquired under JWST Program ID 4212, with principal investigator L.D.B. The datasets generated during and/or analysed during the current study may be obtained from the MAST archive at https://doi.org/10.17909/tcje-1780. All data generated or analysed during this study are included in this Article.

## Code availability

This work made use of NUMPY[61], SCIPY[62], MATPLOTLIB[63] and ASTROPY[64]. SED fit analyses are performed with publicly available software BAGPIPES[9] and PROSPECTOR[10].

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

**Acknowledgements** A.A., E.V., M.M., A.C. and M.R. thank the ISSI for sponsoring the ISSI team: 'Star Formation within rapidly evolving galaxies' in which many ideas discussed in this Article have been brainstormed. A.A. thanks M. Gieles and I. Cabrera-Ziri for their discussions on physical properties of globular clusters. This study is based on the observations with the NASA/ESA/CSA JWST obtained from the Mikulski Archive for Space Telescopes (MAST) at the Space Telescope Science Institute (STScI), which is operated by the Association of Universities for Research in Astronomy (AURA), under NASA contract NAS5-03151. Support for Program number JWST-GO-04212 was provided through a grant from the STScI under NASA contract NAS5-03127. A.A. and A.C. acknowledge support from the Swedish Research Council Vetenskapsrådet (2021-05559). J.M.D. acknowledges the support of project PID2022-138896NB-C51 (MCIU/AEI/MINECO/FEDER, UE) Ministerio de Ciencia, Investigación y Universidades. A.Z. acknowledges support by grant no. 2020750 from the United States–Israel Binational Science Foundation (BSF) and grant no. 2109066 from the United States National Science Foundation (NSF); by the Ministry of Science and Technology, Israel; and by the Israel Science Foundation grant no. 864/23. Y.J.-T. acknowledges financial support from the Horizon 2020 research and innovation programme of the European Union under the Marie Skłodowska–Curie grant agreement no. 898633, the MSCA IF Extensions Program of the Spanish National Research Council (CSIC), the State Agency for Research of the Spanish MCIU through the Center of Excellence Severo Ochoa award to the Instituto de Astrofísica de Andalucía (SEV-2017-0709) and grant CEX2021-001131-S funded by MCIN/AEI/10.13039/501100011033. E.Z. acknowledges project grant 2022-03804 from the Swedish Research Council (Vetenskapsrådet) and has also benefitted from a sabbatical at the Swedish Collegium for Advanced Study. M.O. acknowledges the support from JSPS KAKENHI grant nos. JP22H01260, JP20H05856 and JP22K21349. A.K.I. acknowledges the support from JSPS KAKENHI grant nos. JP23H00131. E.V. and M.M. acknowledge financial support through grants PRIN-MIUR 2020SKSTHZ, the INAF GO grant 2022 'The revolution is around the corner: JWST will probe globular cluster precursors and Population III stellar clusters at cosmic dawn' and by the European Union—NextGenerationEU within PRIN 2022 project no. 20229YBSAN—Globular clusters in cosmological simulations and in lensed fields: from their birth to the present epoch. T.H. is supported by the Leading Initiative for Excellent Young Researchers, MEXT, Japan (HJH02007), and by the JSPS KAKENHI grant no. 22H01258. Y.T. acknowledges the support of the JSPS KAKENHI grant nos. 22H04939 and 23K20035. R.A.W. acknowledges support from NASA JWST Interdisciplinary Scientist grants NAG5-12460, NNX14AN10G and 80NSSC18K0200 from GSFC.

**Author contributions** A.A. performed the analysis of the star clusters and led the discussion and writing of the results, designed the content of the work and superseded its revision. L.D.B. is the principal investigator of the JWST observations and performed the data reduction used in this work. He analysed the host galaxy and oversaw the drafting of the work. E.V. contributed to the analysis, shared the draft writing and co-led the discussion of the results. A.C. contributed to the photometric, SED fitting and magnification extraction analyses. B.W. contributed with the forward model of cluster sizes estimates. J.M.D., G.M., M.O. and K.S. each developed an independent lensing model of the galaxy cluster region. Abdurro'uf produced empirical PSF of the datasets. T.Y.-Y.H., X.X. and A.E.L. contributed to the SED fitting analysis; M.M. estimated the $\beta$ slopes of the star clusters; and E.Z. tested the observed SED against single star models. D.C., M.R. and A.Z. substantially contributed to the discussion and scientific interpretations of the results. G.B. and V.K. helped with data reduction, galaxy photometry and SED analyses in the field of STP-CL J0615-5746, used to constrain the lens models. Y.J.-T. provided independent photometry of the star clusters. S.F., A.K.I., T.R., J.R.R., R.A.W., X.X., T.H. and Y.T. and all the co-authors contributed to the proposal for the observations and aided in the discussion of the results and the content of the paper.

**Funding** Open access funding provided by Stockholm University.

**Competing interests** The authors declare no competing interests.

**Additional information**
**Correspondence and requests for materials** should be addressed to Angela Adamo.

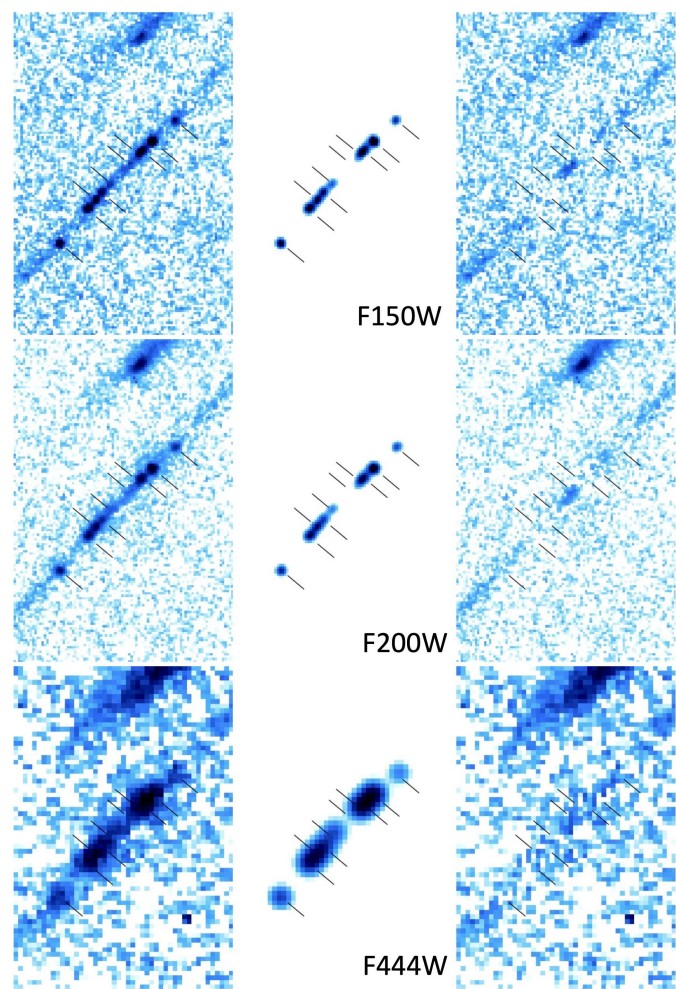

F150W

F200W

F444W

**Extended Data Fig. 1 | Cluster light modelled in the image plane and residuals.** Observed JWST images (left), best-fitted clump shape after removing the local diffuse light, (centre), and residual images (right) in the reference filter F150W (top), and two more bands, the F200W (middle) with similar resolution to the F150W, and the F444W (bottom), with the lowest spatial resolution. We show log-scale images matched in flux in each band.

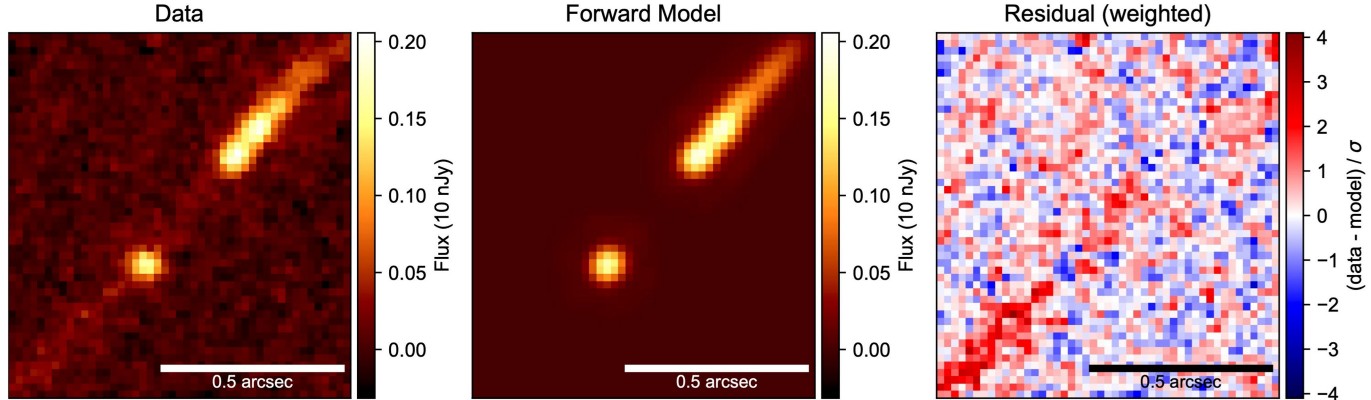

**Extended Data Fig. 2 | Cluster light modelled with forward modelling from the source plane.** Observed JWST image of *Img.1* is shown (far left) along with the best fit image plane from forward modelling (left centre) and weighted fit residuals (right). Weighted residuals are calculated as (data − model)/ (uncertainty).

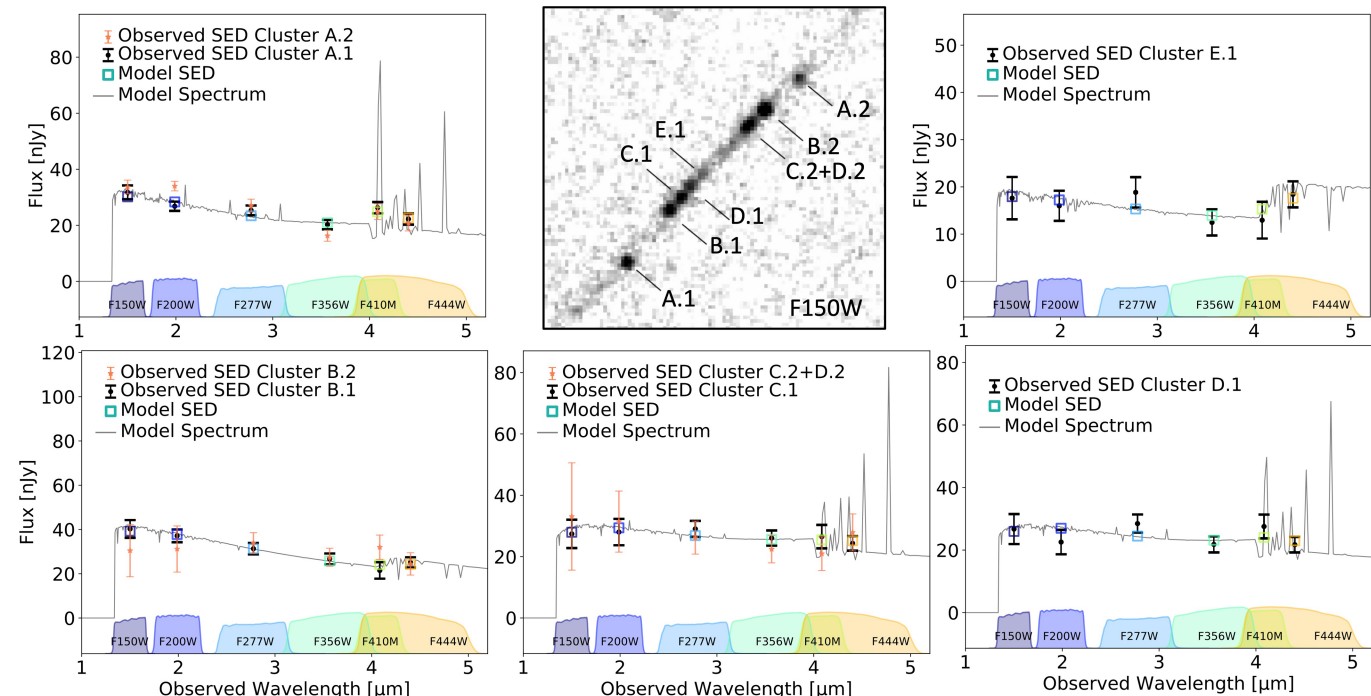

**Extended Data Fig. 3 | Observed photometry and spectral energy distributions (SEDs) of each star cluster (presented in Methods).** We include the observed SEDs of the mirrored image of clusters A, B, C (orange symbols) normalised by the median ratio of the 6 bands, preserving the SED shape.

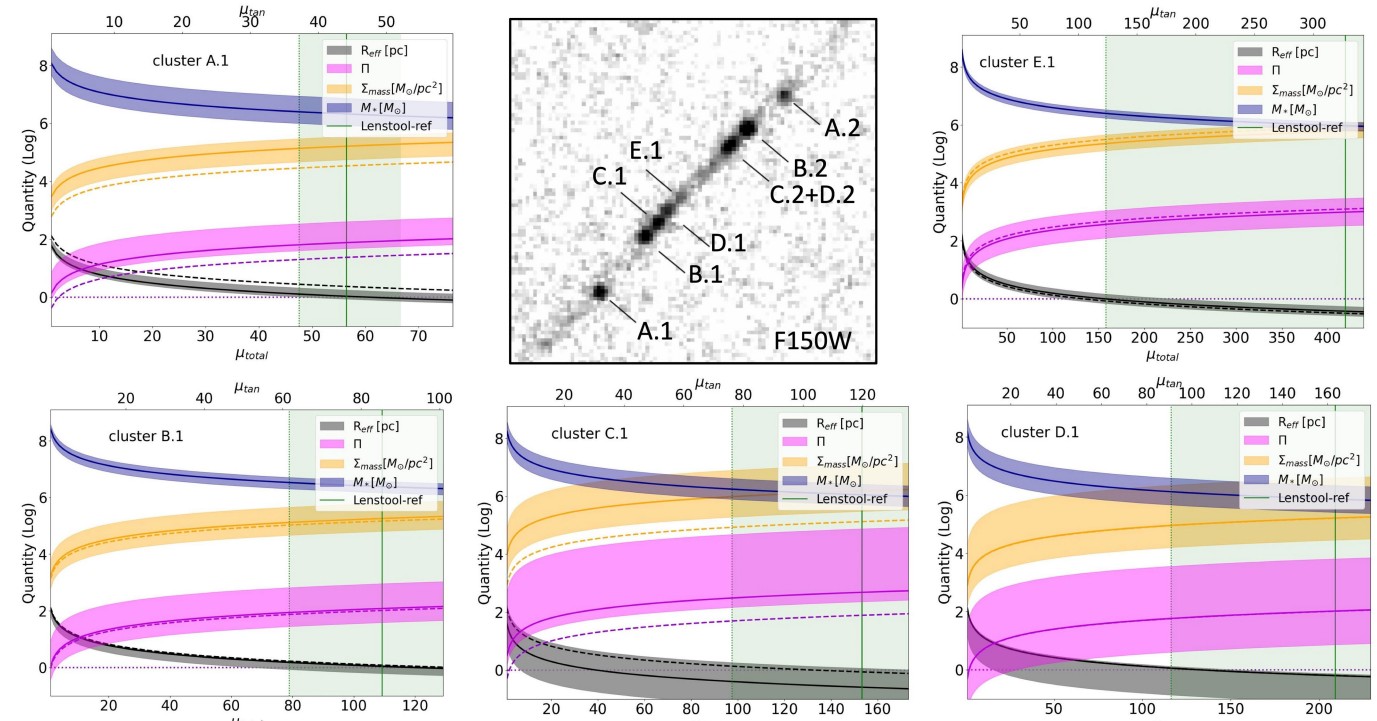

**Extended Data Fig. 4 | Measured and derived cluster physical properties as a function of their magnification.** The most relevant quantities of each cluster in the arc (marked in the central top panel) are expressed as a function of the total magnification ($\mu_{total}$ and in case of the $R_{eff}$ as a function of $\mu_{tan}$). The radii ($R_{eff}$), dynamical ages ($\Pi$), stellar mass surface densities ($\Sigma_{mass}$) and the stellar masses ($M_\star$) suggest the clumps are bound star clusters even at modest magnification regimes ($\mu_{total} > 10$). The transparent green region and the vertical lines show the expected magnification from the reference lens model. The dotted horizontal line indicates the region where $\log(\Pi) = 0$. The shaded areas in the plot mark the uncertainties associated with the derived values. The dashed lines show the lower-limits in $\Sigma_\star$ and $\Pi$, assuming half the stellar PSF FWHM as upper limit for the $R_{eff}$ of each star cluster.

**Extended Data Table 1 | Cluster observed properties**

| ID | F150W | F200W | F277W | F356W | F410M | F444W | $\beta$ |
|---|---|---|---|---|---|---|---|
| A.1 | $27.64 \pm 0.08$ | $27.83 \pm 0.07$ | $27.89 \pm 0.07$ | $28.13 \pm 0.09$ | $27.85 \pm 0.08$ | $28.03 \pm 0.10$ | $-2.36^{+0.19}_{-0.15}$ |
| B.1 | $27.39 \pm 0.11$ | $27.48 \pm 0.08$ | $27.66 \pm 0.09$ | $27.83 \pm 0.10$ | $28.07 \pm 0.19$ | $27.90 \pm 0.10$ | $-2.42^{+0.23}_{-0.19}$ |
| C.1 | $27.81 \pm 0.18$ | $27.78 \pm 0.17$ | $27.74 \pm 0.10$ | $27.86 \pm 0.10$ | $27.84 \pm 0.16$ | $27.94 \pm 0.11$ | $-1.90^{+0.27}_{-0.34}$ |
| D.1 | $27.83 \pm 0.19$ | $28.02 \pm 0.19$ | $27.76 \pm 0.11$ | $28.05 \pm 0.13$ | $27.80 \pm 0.15$ | $28.06 \pm 0.12$ | $-1.80^{+0.35}_{-0.23}$ |
| E.1 | $28.29 \pm 0.28$ | $28.39 \pm 0.22$ | $28.21 \pm 0.19$ | $28.66 \pm 0.24$ | $28.62 \pm 0.33$ | $28.24 \pm 0.16$ | $-1.82^{+0.43}_{-0.46}$ |
| A.2 | $28.08 \pm 0.12$ | $28.07 \pm 0.09$ | $28.32 \pm 0.15$ | $28.87 \pm 0.21$ | $28.41 \pm 0.19$ | $28.60 \pm 0.23$ | $-2.35^{+0.37}_{-0.27}$ |
| B.2 | $27.21 \pm 0.27$ | $27.19 \pm 0.23$ | $27.10 \pm 0.10$ | $27.33 \pm 0.10$ | $27.16 \pm 0.12$ | $27.45 \pm 0.14$ | $-1.80^{+0.31}_{-0.36}$ |
| C.2+D.2 | $27.03 \pm 0.34$ | $27.08 \pm 0.20$ | $27.28 \pm 0.13$ | $27.45 \pm 0.13$ | $27.52 \pm 0.17$ | $27.21 \pm 0.14$ | $-2.45^{+0.46}_{-0.55}$ |

JWST photometry and uncertainties in ABmag and measured $\beta$ slopes of the candidate star clusters.

# Extended Data Table 2 | Compilations of different SED fit outputs described in Methods

| | BAGPIPES $z = 10.2$, $\tau = 1$ Myr, Calzetti attenuation | | | | Prospector SSP, Calzetti attenuation | | | |
|---|---|---|---|---|---|---|---|---|
| ID | Age [Myr] | $10^6 M_*$ [$M_\odot$] | $A_V$ [mag] | $Z/Z_\odot$ [%] | Age [Myr] | $10^6 M_*$ [$M_\odot$] | $A_V$ [mag] | $Z/Z_\odot$ [%] |
| A.1 | $9.2^{+13.5}_{-3.2}$ | $138.34^{+293.88}_{-87.99}$ | $0.22^{+0.15}_{-0.14}$ | $6.38^{+2.42}_{-3.08}$ | $4.0^{+9.5}_{-0.0}$ | $21.94^{+328.83}_{-0.28}$ | $0.00^{+0.27}_{-0.00}$ | $16.91^{+0.87}_{-11.14}$ |
| B.1 | $14.0^{+6.5}_{-4.2}$ | $289.37^{+118.86}_{-137.65}$ | $0.20^{+0.12}_{-0.12}$ | $3.42^{+4.02}_{-2.49}$ | $51.8^{+8.6}_{-21.4}$ | $920.51^{+57.75}_{-238.96}$ | $0.04^{+0.19}_{-0.03}$ | $0.83^{+0.67}_{-0.34}$ |
| C.1 | $9.1^{+9.2}_{-2.8}$ | $172.51^{+271.70}_{-100.24}$ | $0.36^{+0.09}_{-0.16}$ | $4.56^{+3.49}_{-2.88}$ | $14.9^{+23.3}_{-4.7}$ | $585.77^{+268.97}_{-262.10}$ | $0.49^{+0.08}_{-0.29}$ | $0.57^{+1.82}_{-0.41}$ |
| D.1 | $10.9^{+13.1}_{-3.9}$ | $235.49^{+256.24}_{-154.93}$ | $0.32^{+0.12}_{-0.16}$ | $5.27^{+3.12}_{-3.03}$ | $16.2^{+19.6}_{-5.6}$ | $528.51^{+327.14}_{-168.52}$ | $0.51^{+0.07}_{-0.27}$ | $0.45^{+2.17}_{-0.29}$ |
| E.1 | $36.8^{+19.7}_{-16.4}$ | $421.73^{+153.99}_{-151.94}$ | $0.21^{+0.18}_{-0.15}$ | $5.05^{+3.15}_{-3.14}$ | $51.8^{+29.9}_{-24.2}$ | $582.23^{+254.82}_{-266.12}$ | $0.11^{+0.27}_{-0.09}$ | $2.66^{+9.38}_{-2.08}$ |
| A.2 | $17.2^{+8.8}_{-6.2}$ | $166.70^{+90.28}_{-77.77}$ | $0.11^{+0.11}_{-0.07}$ | $4.39^{+3.86}_{-3.28}$ | $8.0^{+18.7}_{-0.1}$ | $45.26^{+278.14}_{-1.60}$ | $0.01^{+0.23}_{-0.01}$ | $19.27^{+0.67}_{-15.80}$ |
| B.2 | $9.3^{+6.0}_{-2.6}$ | $294.46^{+314.14}_{-157.81}$ | $0.28^{+0.13}_{-0.17}$ | $4.83^{+3.41}_{-2.89}$ | $13.6^{+12.1}_{-3.1}$ | $743.34^{+181.02}_{-256.28}$ | $0.40^{+0.15}_{-0.27}$ | $1.24^{+3.65}_{-1.02}$ |
| C.2+D.2 | $18.6^{+7.8}_{-7.4}$ | $603.09^{+135.33}_{-263.11}$ | $0.19^{+0.16}_{-0.13}$ | $6.29^{+2.63}_{-3.50}$ | $24.5^{+11.0}_{-11.0}$ | $794.45^{+136.02}_{-308.30}$ | $0.13^{+0.27}_{-0.11}$ | $3.71^{+6.39}_{-3.25}$ |

| | BAGPIPES $z = 10.2$, Burst, Calzetti attenuation | | | | BAGPIPES $z = 10.2$, BPASS, $\tau = 1$ Myr, Calzetti attenuation | | | |
|---|---|---|---|---|---|---|---|---|
| A.1 | $23.5^{+4.2}_{-17.0}$ | $503.90^{+106.02}_{-409.27}$ | $0.32^{+0.09}_{-0.17}$ | $0.74^{+4.93}_{-0.70}$ | $16.68^{+10.41}_{-7.65}$ | $257.14^{+159.40}_{-171.21}$ | $0.09^{+0.10}_{-0.06}$ | $3.33^{+3.66}_{-2.19}$ |
| B.1 | $15.4^{+6.0}_{-6.6}$ | $421.94^{+129.62}_{-229.56}$ | $0.31^{+0.10}_{-0.17}$ | $0.17^{+0.75}_{-0.14}$ | $18.05^{+8.39}_{-5.67}$ | $319.09^{+153.12}_{-133.81}$ | $0.09^{+0.08}_{-0.06}$ | $0.63^{+0.90}_{-0.42}$ |
| C.1 | $7.0^{+13.8}_{-4.7}$ | $149.70^{+434.40}_{-99.12}$ | $0.37^{+0.09}_{-0.20}$ | $0.30^{+2.71}_{-0.27}$ | $14.32^{+14.20}_{-11.11}$ | $269.70^{+275.99}_{-236.38}$ | $0.17^{+0.15}_{-0.11}$ | $1.69^{+2.63}_{-1.09}$ |
| D.1 | $8.4^{+16.0}_{-3.7}$ | $173.50^{+391.91}_{-116.78}$ | $0.36^{+0.10}_{-0.18}$ | $0.57^{+3.84}_{-0.53}$ | $20.70^{+10.67}_{-8.72}$ | $373.07^{+166.15}_{-196.33}$ | $0.20^{+0.15}_{-0.12}$ | $1.94^{+3.61}_{-1.31}$ |
| E.1 | $41.0^{+24.2}_{-14.4}$ | $439.84^{+133.47}_{-112.90}$ | $0.22^{+0.17}_{-0.14}$ | $0.22^{+2.52}_{-0.19}$ | $31.75^{+22.97}_{-14.31}$ | $324.68^{+158.77}_{-127.03}$ | $0.17^{+0.16}_{-0.11}$ | $3.66^{+4.00}_{-2.55}$ |
| A.2 | $18.6^{+8.0}_{-6.3}$ | $222.91^{+73.55}_{-74.71}$ | $0.19^{+0.15}_{-0.11}$ | $0.28^{+2.60}_{-0.24}$ | $18.49^{+9.68}_{-6.37}$ | $171.49^{+99.97}_{-78.39}$ | $0.06^{+0.08}_{-0.04}$ | $1.33^{+2.20}_{-0.91}$ |
| B.2 | $7.7^{+4.3}_{-2.3}$ | $263.42^{+302.06}_{-125.30}$ | $0.33^{+0.11}_{-0.16}$ | $1.04^{+3.22}_{-0.98}$ | $12.30^{+8.04}_{-8.31}$ | $382.18^{+286.08}_{-324.66}$ | $0.14^{+0.13}_{-0.10}$ | $2.04^{+3.46}_{-1.37}$ |
| C.2+D2 | $19.9^{+7.3}_{-10.0}$ | $670.79^{+100.62}_{-317.92}$ | $0.19^{+0.17}_{-0.14}$ | $0.99^{+3.98}_{-0.93}$ | $19.18^{+8.00}_{-7.78}$ | $521.93^{+201.93}_{-248.80}$ | $0.11^{+0.13}_{-0.08}$ | $3.02^{+4.10}_{-1.77}$ |

| | BAGPIPES $z = 10.2$, $\tau = 1$ Myr, SMC extinction | | | | BAGPIPES $z = 10.2$, $\tau =$ free, Calzetti attenuation | | | | |
|---|---|---|---|---|---|---|---|---|---|
| ID | Age [Myr] | $10^6 M_*$ [$M_\odot$] | $A_V$ [mag] | $Z/Z_\odot$ [%] | Age [Myr] | $10^6 M_*$ [$M_\odot$] | $A_V$ [mag] | $Z/Z_\odot$ [%] | $\tau$ [Myr] |
| A.1 | $19.3^{+12.9}_{-13.4}$ | $278.55^{+166.88}_{-234.09}$ | $0.04^{+0.04}_{-0.03}$ | $6.67^{+2.39}_{-3.60}$ | $27.90^{+21.74}_{-14.82}$ | $192.02^{+135.05}_{-147.57}$ | $0.08^{+0.08}_{-0.05}$ | $6.39^{+2.40}_{-3.13}$ | 60.79 |
| B.1 | $16.4^{+6.3}_{-5.7}$ | $272.67^{+114.16}_{-115.28}$ | $0.05^{+0.03}_{-0.03}$ | $4.62^{+3.02}_{-3.17}$ | $19.15^{+14.53}_{-7.31}$ | $162.58^{+97.53}_{-5.19}$ | $0.04^{+0.06}_{-0.03}$ | $2.16^{+2.54}_{-1.55}$ | 65.22 |
| C.1 | $12.6^{+8.2}_{-4.8}$ | $259.80^{+160.50}_{-157.52}$ | $0.16^{+0.06}_{-0.07}$ | $4.10^{+3.46}_{-2.76}$ | $15.68^{+16.10}_{-9.40}$ | $171.97^{+137.98}_{-69.69}$ | $0.20^{+0.12}_{-0.11}$ | $3.84^{+3.36}_{-2.33}$ | 60.52 |
| D.1 | $13.6^{+10.7}_{-6.0}$ | $249.39^{+179.19}_{-112.66}$ | $0.15^{+0.07}_{-0.07}$ | $4.53^{+3.60}_{-3.00}$ | $18.39^{+22.90}_{-10.93}$ | $167.89^{+176.63}_{-86.57}$ | $0.19^{+0.13}_{-0.11}$ | $5.08^{+2.97}_{-3.12}$ | 69.27 |
| E.1 | $35.3^{+17.9}_{-14.6}$ | $357.91^{+148.91}_{-112.66}$ | $0.11^{+0.08}_{-0.08}$ | $5.38^{+3.27}_{-3.66}$ | $54.85^{+42.68}_{-30.97}$ | $314.15^{+176.71}_{-68.91}$ | $0.19^{+0.15}_{-0.13}$ | $4.15^{+3.79}_{-2.70}$ | 67.05 |
| A.2 | $18.3^{+7.8}_{-6.6}$ | $157.81^{+78.97}_{-69.06}$ | $0.04^{+0.04}_{-0.02}$ | $4.02^{+3.60}_{-2.76}$ | $27.56^{+22.23}_{-13.42}$ | $123.81^{+97.75}_{-35.06}$ | $0.05^{+0.07}_{-0.03}$ | $3.63^{+3.32}_{-2.34}$ | 65.64 |
| B.2 | $11.1^{+6.6}_{-3.5}$ | $350.70^{+271.40}_{-194.18}$ | $0.14^{+0.07}_{-0.07}$ | $4.44^{+3.83}_{-2.76}$ | $15.87^{+15.57}_{-7.76}$ | $270.99^{+209.38}_{-114.48}$ | $0.16^{+0.12}_{-0.10}$ | $4.43^{+3.55}_{-2.59}$ | 64.48 |
| C.2+D2 | $22.6^{+6.2}_{-7.3}$ | $632.09^{+120.30}_{-206.55}$ | $0.08^{+0.08}_{-0.06}$ | $5.56^{+2.86}_{-3.05}$ | $34.25^{+16.40}_{-19.41}$ | $474.27^{+173.05}_{-48.73}$ | $0.13^{+0.15}_{-0.09}$ | $3.79^{+3.77}_{-2.60}$ | 69.46 |

The reported masses are not corrected for magnification. The output of the fit assuming τ=1 Myr and Calzetti attenuation is referred to as BAGPIPES-*exp* and used to derive the physical values reported in Table 1. For the BAGPIPES fit with τ=free we report mass weighted quantities.

**Extended Data Table 3 | Magnifications and associated uncertainties for 3 different lens models**

| ID | RA | DEC | $\mu_{\mathrm{Lenstool-A}}$ $(\mu_{\mathrm{tan}})$ | $\mu_{\mathrm{Lenstool-B}}$ $(\mu_{\mathrm{tan}})$ | $\mu_{\mathrm{glafic}}$ $(\mu_{\mathrm{tan}})$ |
|---|---|---|---|---|---|
| A.1 | 93.979828 | -57.772475 | $56.5^{+10.1}_{-8.9}(44.8)$ | $122.0^{+48.8}_{-17.8}(81.9)$ | $76.9\ (68.2)$ |
| B.1 | 93.979698 | -57.772395 | $109.2^{+31.2}_{-30.3}(84.8)$ | $212.6^{+152.8}_{-25.4}(142.3)$ | $153.8\ (125.7)$ |
| C.1 | 93.979660 | -57.772373 | $153.3^{+91.9}_{-55.8}(120.8)$ | $280.0^{+270.7}_{-27.2}(187.9)$ | $200.9\ (177.5)$ |
| D.1 | 93.979635 | -57.772355 | $209.1^{+132.8}_{-93.0}(159.8)$ | $349.5^{+474.6}_{-56.5}(234.2)$ | $288.8\ (254.7)$ |
| E.1 | 93.979599 | -57.772335 | $419.3^{+215.7}_{-261.6}(341.2)$ | $527.3^{+708.6}_{-30.1}(353.7)$ | $516.4\ (455.7)$ |
| A.2 | 93.979316 | -57.772180 | $57.7^{+27.3}_{-3.7}(46.5)$ | $128.9^{+51.6}_{-17.1}(89.5)$ | $76.3\ (70.1)$ |
| B.2 | 93.979416 | -57.772235 | $97.8^{+63.5}_{-10.0}(72.0)$ | $234.3^{+188.7}_{-30.8}(157)$ | $133.0\ (116.9)$ |
| C.2+D.2 | 93.979460 | -57.772256 | $138.4^{+263.7}_{-20.4}\ (117)$ | $372.8^{+527.0}_{-58.2}(249.6)$ | $207.7\ (182.3)$ |

The model *Lenstool-A* is used as reference in the analysis.