## [Peer Review File · Nature]

Manuscript Title: Bound star clusters observed in a lensed galaxy 460 Myr after the Big Bang

Reviewer Comments & Author Rebuttals

Reviewer Reports on the Initial Version:

Referees' comments:

Referee #1 (Remarks to the Author):

I have read and reviewed the manuscript titled "The discovery of bound star clusters 460 Myr after the Big Bang" with great interest. It presents the discovery of gravitationally bound star-cluster structures at redshift 10.2, the most distant such measurement to date, using JWST observations and gravitational lensing. These early star clusters represent the progenitors of present-day globular clusters and appear to be even more compact and dense. The discovery is made possible by the magnification induced by the gravitational lensing effect, allowing to spatially resolve structures down to the pc-level.

The study and results presented in this work are a unique, compelling, and timely discovery, well worthy of publication in Nature. That being said, I have several major comments about the redshift measurements and SED-modeling procedures that I would like the authors to address before I can recommend the manuscript for publication.

Crucial Comment:

1. I am most strongly concerned about the redshift measurement: It is stated in the manuscript that this study is based on a photometric redshift measurement from the JWST/NIRCam broad-band photometry and not on a spectroscopic redshift measurement. Broad-band photometric redshifts are however prone to large uncertainties and systematics due to SED-modeling assumptions and the very 'crude' sampling of the SED provided by broad-band filters. This is in particular the case for JWST-detected $z > 10$ object candidates, which present strong photometric-redshift degeneracies (e.g. Furtak et al. 2023) and have often subsequently been revealed as much lower-redshift objects (e.g. Zavala et al. 2022). While the authors state in line 142 that their photometric modeling "excludes lower-redshift solutions", this analysis is not shown or even discussed in the paper. I understand that the authors intend to present the details of this analysis in a subsequent companion paper, which is however not yet published (or even posted as a pre-print), meaning that it is impossible for the reader to verify and understand the redshift measurement. Since the entire analysis and results presented here depend on the galaxy indeed being at $z=10$, this is unfortunately too important to not be addressed here and needs to be clarified before I can recommend the manuscript for publication.

In addition, I see in the public records of StScI that the JWST program 4212, on which this work is based, also includes a NIRSpec spectroscopic observation. I am therefore wondering why the spectroscopic data are not used in this work.

In light of all that, I very strongly recommend that the authors measure the redshift and physical parameters of the bound star clusters presented here directly from the NIRSpec spectroscopy. Such a measurement would eliminate any possible doubt as to the results and would amply merit publication in Nature.

If that is for some reason not possible, I would recommend that the authors detail and thoroughly discuss the photometric redshift analysis in this manuscript.

Major comments:

2. I have several comments regarding the SED-modeling analysis performed in this work

a) Since SED-modeling broad-band photometry is prone to some systematics and degeneracies, did the authors verify their results by running a second code in addition to BAGPIPES, e.g. prospector? This would greatly improve the robustness of the results.

b) Were the fluxes de-magnified before SED-fitting (and the uncertainties propagated)? Recent studies with physically motivated codes such as BAGPIPES, prospector or BEAGLE have been shown to yield different results between modeling magnified and de-magnified fluxes, in particular in the high-magnification regime. In addition, the latest version of prospector even includes the magnification into the modeling process (see e.g. Wang et al. 2024).

c) The SED-fitting procedure assumes a Calzetti dust attenuation law. At high redshifts and low metallicities, the SMC law has however also been found to fit well and even better sometimes. Could the authors therefore comment on their choice and perhaps (briefly) quantify how the SED-fitting derived quantities change with the other attenuation law?

d) In lines 284-285, the authors state that the dust attenuations and metallicities of the bound star clusters agree with the galactic measurement of the host arc. However, comparing the values in Table 1 and the Cosmic Gems arc values in line 228, the star clusters' A_v values seem to be significantly higher than the host galaxy's, by about a factor 10. Is this expected for star clusters compared to their host galaxy?

e) In line 497, the redshift is fixed to 10.2 for the SED-modeling analysis. Given that this is a photometric redshift, with significant uncertainties of 0.2 and perhaps even larger systematic uncertainty, it would be useful to verify how the physical parameters of the star clusters depend on the redshift assumed for the fitting. This can be done by either leaving the redshift as a completely free parameter or by placing a Gaussian prior corresponding to the photometric redshift estimate of the host galaxy on it. The latter would in addition have the benefit to fold the redshift uncertainty into the physical parameter uncertainties.

3. In line 470 the authors describe how the uncertainties on photometry and size were derived. Were the magnification uncertainties propagated into these quantities? If yes, did these include the lens modeling systematics?

Minor comments:

4. In line 111: Perhaps also cite the recent work by Atek et al. 2023 here for the ionization power of young low-mass galaxies in the EoR (also again later, in line 382).

5. In line 180: For MACS0647-JD, there was a paper presenting the spectroscopic redshift measurement in addition to the photometric paper cited here (reference number 8), Hsiao et al. 2023b.

6. In Table 1: I am aware that the main text specifies that the stellar masses reported here are de-magnified, but I would nevertheless recommend also specifying this in the table or its caption, to make sure it is immediately visible and avoid possible confusion.

7. In line 399: Double 'the'.

8. In line 444, it is stated that the tangential magnifications were used for the radius measurements, the tables however report the total and the radial magnifications. While I understand the reason why this is done, I would nevertheless suggest to perhaps simply report the total and tangential magnifications in each table, to avoid confusion and be more clear to readers who are not necessarily experts on lensing.

Referee #2 (Remarks to the Author):

Please incorporate the points below into your comments to the Authors. For all other article types, such as review or progress articles, please provide comments to authors and editors in the boxes provided below.

Summary of the key results

The paper reports the discovery of young massive star clusters in a strongly lensed galaxy at redshift ~ 10.2 , when the Universe was only 420 Myr old.

The observations are from JWST/NIRCam and reveal five distinct clusters threaded along the arc, which drop out at F115W.

The clusters have a de-lensed size of ~ 1 pc, making up 60% of total FUV light.

They are fairly young (< 35 Myr), and are fairly massive for the era ($10^6 M_{\text{sun}}$), resulting in extremely high surface densities, three orders of magnitude above local universe star clusters.

These are the earliest known proto-globular cluster candidates.

Originality and significance: if not novel, please include references.

If the redshift is confirmed spectroscopically, this will be the most magnified object at the highest redshift, resolved to the parsec scale. Based on the object's dropout in F115W, the photometric redshift is $z > 9$. Hence, this is a novel and significant observational result.

Data & methodology: validity of approach, quality of data, quality of presentation.

Since most photometric and size measurement methodologies will be described in a paper in preparation, reviewing the analysis techniques has been challenging. I am requesting the latest draft of Bradley et al. in prep to gauge the data quality and analysis fully.

The observation is novel enough for this paper to merit publication. To increase the robustness of the reported physical parameters, I suggest the following improvements:

It is not trustworthy to measure sizes smaller than the PSF's HWHM. Many of the sizes quoted in Table 1 are below this limit. I am requesting a simulation demonstrating their technique's size and flux limit both in the image and source planes.

The sizes and, consequently, the fluxes are measured by modelling a point spread function or the shape of the clumps in extremely small cutouts (7x7 or 11x11 pix). Such a small cutout cannot fit the wings of the PSF, nor will it be able to estimate the background properly, separating it from the light of the arc. I am requesting a simulation demonstrating the effectiveness of their technique's total light recovery. The uncertainty in flux measurement from a small cutout will likely be very high, depending on the background level and will vary with wavelength.

Even though three sets of SED fitting have been performed, the assumptions on SFH are not varied enough to probe the range of possible ages. The exponential mode assumes a very short $\tau = 1$ Myr, essentially equivalent to the burst model. This is reflected in the similar physical properties recovered from these two fits. I suggest allowing for multiple bursts and longer taus to probe possibilities of older age.

The SED fitting has been performed with fixed assumptions on ionizing parameters, narrow SFH, IMF, etc. While the current photometry will not be able to reveal any new information on these parameters, changing the fixed value will give very different stellar population properties. I am interested in how these clusters' age and stellar mass change due to different IMF-alpha (indications of top-heavy IMF at these redshifts), ionization parameters, and broader SFH.

Have the authors tried fitting with other SED methods such as Prospector, Cigale, and Dense Basis? How do the stellar population properties change when non-parametric SFH are assumed?

Appropriate use of statistics and treatment of uncertainties.

While it has been challenging to get a full picture of the analysis since many of the details will be presented in a paper in preparation, I believe many of the uncertainties on derived stellar properties will increase when they consider the suggestions presented in the previous section.

Conclusions: robustness, validity, reliability

The observation is reliable, but the photometry and SED fitting techniques are not robust (see section 3).

Suggested improvements: experiments, data for possible revision

Please see section 3 for suggested improvements.

References: appropriate credit to previous work?

The references seem adequate.

Clarity and context: lucidity of abstract/summary, appropriateness of abstract, introduction and conclusions.

The paper seems hastily written and is missing a discussion on the implications of finding these star clusters at $z > 10$ on globular cluster formation and the assembly of galaxies. The observation is extremely exciting, but the paper reads very dryly and will not be of interest to someone who is not in this field.

Please incorporate the points below into your comments to the Authors. For all other article types, such as review or progress articles, please provide comments to authors and editors in the boxes provided below.

1. Summary of the key results

1. The paper reports the discovery of young massive star clusters in a strongly lensed galaxy at redshift ~ 10.2 , when the Universe was only 420 Myr old.
2. The observations are from JWST/NIRCam and reveal five distinct clusters threaded along the arc, which drop out at F115W.
3. The clusters have a de-lensed size of ~ 1 pc, making up 60% of total FUV light.
4. They are fairly young (< 35 Myr), and are fairly massive for the era (10^6 Msun), resulting in extremely high surface densities, three orders of magnitude above local universe star clusters.
5. These are the earliest known proto-globular cluster candidates.

2. Originality and significance: if not novel, please include references.

If the redshift is confirmed spectroscopically, this will be the most magnified object at the highest redshift, resolved to the parsec scale. Based on the object's dropout in F115W, the photometric redshift is $z > 9$. Hence, this is a novel and significant observational result.

3. Data & methodology: validity of approach, quality of data, quality of presentation.

Since most photometric and size measurement methodologies will be described in a paper in preparation, reviewing the analysis techniques has been challenging. I am requesting the latest draft of Bradley et al. in prep to gauge the data quality and analysis fully.

The observation is novel enough for this paper to merit publication. To increase the robustness of the reported physical parameters, I suggest the following improvements:

1. It is not trustworthy to measure sizes smaller than the PSF's HWHM. Many of the sizes quoted in Table 1 are below this limit. I am requesting a simulation demonstrating their technique's size and flux limit both in the image and source planes.
2. The sizes and, consequently, the fluxes are measured by modelling a point spread function or the shape of the clumps in extremely small cutouts (7x7 or 11x11 pix). Such a small cutout cannot fit the wings of the PSF, nor will it be able to estimate the background properly, separating it from the light of the arc. I am requesting a simulation demonstrating the effectiveness of their technique's total light recovery. The uncertainty in flux measurement from a small cutout will likely be very high, depending on the background level and will vary with wavelength.
3. Even though three sets of SED fitting have been performed, the assumptions on SFH are not varied enough to probe the range of possible ages. The exponential mode assumes a very short $\tau = 1$ Myr, essentially equivalent to the burst model. This is reflected in the similar physical properties recovered

from these two fits. I suggest allowing for multiple bursts and longer taus to probe possibilities of older age.

4. The SED fitting has been performed with fixed assumptions on ionizing parameters, narrow SFH, IMF, etc. While the current photometry will not be able to reveal any new information on these parameters, changing the fixed value will give very different stellar population properties. I am interested in how these clusters' age and stellar mass change due to different IMF-alpha (indications of top-heavy IMF at these redshifts), ionization parameters, and broader SFH.
5. Have the authors tried fitting with other SED methods such as Prospector, Cigale, and Dense Basis? How do the stellar population properties change when non-parametric SFH are assumed?

4. Appropriate use of statistics and treatment of uncertainties.

While it has been challenging to get a full picture of the analysis since many of the details will be presented in a paper in preparation, I believe many of the uncertainties on derived stellar properties will increase when they consider the suggestions presented in the previous section.

5. Conclusions: robustness, validity, reliability

The observation is reliable, but the photometry and SED fitting techniques are not robust (see section 3).

6. Suggested improvements: experiments, data for possible revision

Please see section 3 for suggested improvements.

7. References: appropriate credit to previous work?

The references seem adequate.

8. Clarity and context: lucidity of abstract/summary, appropriateness of abstract, introduction and conclusions.

The paper seems hastily written and is missing a discussion on the implications of finding these star clusters at $z > 10$ on globular cluster formation and the assembly of galaxies. The observation is extremely exciting, but the paper reads very dryly and will not be of interest to someone who is not in this field.

Author Rebuttals to Initial Comments:

Science revision

1. Referee 1 checked the observing records and finds that you already have a spectrum, which you should present in the paper, to remove all doubt about the redshift. From a personal perspective, I would find it pointless to publish a photometric redshift when you already have a spectrum.

>>>We have addressed this comment below. We would like to point out that a Cycle3 spectroscopic program has been accepted and will provide definitive redshift.

2. Referee 2 notes that the paper seems hastily written, and phrased in a dry way that would make the paper of little interest to someone not in the field: we agree. Instead of discussing the implications, the final paragraph of the main text is essentially just advertising (there is a fair bit of that through the paper, and it all must be removed).

>>>We have rewritten the final paragraphs to discuss possible implications of the results.

Referees' comments:

Referee #1 (Remarks to the Author):

I have read and reviewed the manuscript titled "The discovery of bound star clusters 460 Myr after the Big Bang" with great interest. It presents the discovery of gravitationally bound star-cluster structures at redshift 10.2, the most distant such measurement to date, using JWST observations and gravitational lensing. These early star cluster represent the progenitors of present-day globular clusters and appear to be even more compact and dense. The discovery is made possible by the magnification induced by the gravitational lensing effect, allowing to spatially resolve structures down to the pc-level.

The study and results presented in this work are a unique, compelling, and timely discovery, well worthy of publication in Nature. That being said, I have several major comments about the redshift measurements and SED-modeling procedures that I would like the authors to address before I can recommend the manuscript for publication.

Crucial Comment:

1. I am most strongly concerned about the redshift measurement: It is stated in the manuscript that this study is based on a photometric redshift measurement from the JWST/NIRCam broad-band photometry and not on a spectroscopic redshift measurement. Broad-band photometric redshifts are however prone to large uncertainties and systematics due to SED-modeling assumptions and the very 'crude' sampling of the SED provided by broad-band filters. This is in particular the case for JWST-detected $z > 10$ object candidates, which present strong photometric-redshift degeneracies (e.g. Furtak et al. 2023) and have often subsequently been revealed as much lower-redshift objects (e.g. Zavala et al. 2022). While the authors state in line 142 that their photometric modeling "excludes lower-redshift solutions", this analysis is not shown or even discussed in the paper. I understand that the authors intend to present the details of this analysis in a subsequent companion paper, which is however not yet published (or even posted as a pre-print), meaning that it is impossible for the reader to verify and understand the redshift measurement. Since the entire analysis and results presented here depend on the galaxy indeed being at $z = 10$, this is unfortunately too important to not be addressed here and needs to be clarified before I can recommend the manuscript for publication.

In addition, I see in the public records of StScI that the JWST program 4212, on which this work is based, also includes a NIRSpec spectroscopic observation. I am therefore wondering why the spectroscopic data are not used in this work.

In light of all that, I very strongly recommend that the authors measure the redshift and physical parameters of the bound star clusters presented here directly from the NIRSpec spectroscopy. Such a measurement would eliminate any possible doubt as to the results and would amply merit publication in Nature.

If that is for some reason not possible, I would recommend that the authors detail and thoroughly discuss the photometric redshift analysis in this manuscript.

>>>>We agree with the points raised by the referee, a strict redshift determination is fundamental. Indeed the JWST program 4212 included NIRSpec G395H high-resolution MSA spectroscopy, which is supposed to cover the NeIII and OII and other faint recombination lines in the NUV. The spectroscopy unfortunately does not reveal any strong emission, which could aid in the redshift determination. In total 3 shutters cover the entire region containing the star clusters analyzed here in image 1 and 2 of the arc. The program was high-risk high-gain and based on the information available at the time with HST imaging (see below F140W and F160W), e.g. 2 compact clumps. In cycle 2, we opted for a setup that could allow us to derive ISM physical properties. When the JWST imaging became available, we could see that the clumps are actually multiple objects. This became a limitation in terms of the spectroscopy as sensitivity estimates were made for an unresolved compact object, which might explain the lack of detection of line emissions. We got cycle 3 observing time to get NIRSpec prism observation of the Ly α break, a well known and robust technique to confirm the redshift (e.g. Curtis-Lake et al. 2023). In support of this manuscript, we provide some extracts of the Bradley et al in prep paper, which we hope the referee can keep confidential. The Cosmic Gems arc is a JWST/F115W dropout (see footprints including HST and JWST, the latter highlighted in orange). The redshift determination is done including HST and JWST combined photometry. It is a narrow redshift 10.2 ± 0.2 (95% confidence level). The lower redshift solution is a poor fit to the observed data and would be discarded by spectroscopy at hand as it should have optical strong line emissions.

Major comments:

2. I have several comments regarding the SED-modeling analysis performed in this work

a) Since SED-modeling broad-band photometry is prone to some systematics and degeneracies, did the authors verify their results by running a second code in addition to BAGPIPES, e.g. prospector? This would greatly improve the robustness of the results.

>>> We have added in the Method section and Table 3 the outputs obtained with Prospector. In that case we used SSP assumptions, as that feature is not available for BAGPIPES. The results are consistent, within uncertainties. We have added a paragraph in Methods to describe the fit done with prospector and how they impact the results presented in the manuscript. We notice that we have included several other tests per request of reviewer #2.

b) Were the fluxes de-magnified before SED-fitting (and the uncertainties propagated)? Recent studies with physically motivated codes such as BAGPIPES, prospector or BEAGLE have been shown to yield different results between modeling magnified and de-magnified fluxes, in particular in the high-magnification regime. In addition, the latest version of prospector even includes the magnification into the modeling process (see e.g. Wang et al. 2024).

>>>> The SED fitting is done on fluxes that are not de-magnified, this is true for both the star clusters and the galaxy. We then apply the same magnification value to the resulting mass, which is like applying the same magnification to each fitted flux, as lensing is achromatic. Star clusters are very compact, but we take into account the magnification variations at their position, using the minimum and major axes of the fitted shape (e.g. see Claeysens et al 2023 for a careful description of how magnification and uncertainties are extracted at the position of the clusters). The referee is right that for extended objects such as the Cosmic Gems arc, this might not be fully correct because of the variations in magnifications across the body of the system. We use the method typically adopted for lensing studies (e.g., Vanzella et al 2023b). We fit the SED of the galaxy without correcting for magnification, we then apply the median value of magnification extracted in the same ellipse the photometry has been performed, which has been shown to be more robust than the mean (the latter is skewed towards high values because of the proximity to the critical curve). We mention in the text that the mass of the galaxy is less robust than using the FUV light. In that case we do not need to apply any magnification to cluster and galaxy light, but compare the two directly.

c) The SED-fitting procedure assumes a Calzetti dust attenuation law. At high redshifts and low metallicities, the SMC law has however also been found to fit well and even better sometimes. Could the authors therefore comment on their choice and perhaps (briefly) quantify how the SED-fitting derived quantities change with the other attenuation law?

>>>> To address this point we have run BAGPIPES assuming a $\tau=1$ Myr exponential decline but using a SMC extinction law, we include here below a table summarizing the recovered physical parameters. Overall the values are in very good agreement within 68% confidence level in the two different sets of fits. We notice that the recovered extinction is lower for the run with the SMC extinction model, as expected due to the intrinsic differences between Calzetti vs. SMC. We have added a sentence in the method section that reports the outcome of this exercise and the outputs are listed in table 3 of the Extended Data section.

Table 5 Comparisons between SED fit outputs estimated with BAGPIPES $\tau=1$ Myr, $z=10.2$, & Calzetti attenuation law vs. BAGPIPES $\tau=1$ Myr, $z=10.2$, & SMC extinction law. The reported masses are not corrected for magnification.

ID	Magnification models		BAGPIPES $\tau=1$ Myr, $z=10.2$, & Calzetti				BAGPIPES $\tau=1$ Myr, $z=10.2$, & SMC			
	$\mu_{\text{lenstool-B}} (\mu_r)$	$\mu_{\text{glafic}} (\mu_r)$	Age [Myr]	$10^6 M_* [M_\odot]$	A_V [mag]	Z/Z_\odot [%]	Age [Myr]	$10^6 M_* [M_\odot]$	A_V [mag]	Z/Z_\odot [%]
A.1	122.0 ^{+48.8} _{-17.8}	76.9	9.2 ^{+13.5} _{-3.2}	138.34 ^{+293.88} _{-87.99}	0.22 ^{+0.15} _{-0.14}	6.38 ^{+2.42} _{-3.08}	19.32 ^{+12.89} _{-13.39}	278.55 ^{+166.88} _{-234.09}	0.04 ^{+0.04} _{-0.03}	6.67 ^{+2.39} _{-3.60}
B.1	212.6 ^{+152.8} _{-25.4}	153.8	14.0 ^{+6.5} _{-4.2}	289.37 ^{+118.86} _{-137.65}	0.20 ^{+0.12} _{-0.12}	3.42 ^{+4.02} _{-2.49}	16.37 ^{+6.34} _{-5.74}	272.67 ^{+114.16} _{-115.28}	0.05 ^{+0.03} _{-0.03}	4.62 ^{+3.02} _{-3.17}
C.1	280.0 ^{+270.7} _{-27.2}	200.9	9.1 ^{+9.2} _{-2.8}	172.51 ^{+231.90} _{-100.24}	0.36 ^{+0.05} _{-0.16}	4.56 ^{+3.49} _{-2.88}	12.59 ^{+8.24} _{-4.80}	259.80 ^{+160.50} _{-157.52}	0.16 ^{+0.06} _{-0.07}	4.10 ^{+3.46} _{-2.76}
D.1	349.5 ^{+474.6} _{-56.5}	288.8	10.9 ^{+13.1} _{-3.9}	235.49 ^{+256.24} _{-154.93}	0.32 ^{+0.12} _{-0.16}	5.27 ^{+3.12} _{-3.03}	13.62 ^{+10.68} _{-6.01}	249.39 ^{+179.19} _{-168.07}	0.15 ^{+0.07} _{-0.07}	4.53 ^{+3.60} _{-3.00}
E.1	527.3 ^{+708.6} _{-30.1}	516.4	36.8 ^{+19.7} _{-16.4}	421.73 ^{+153.99} _{-90.28}	0.21 ^{+0.18} _{-0.15}	5.05 ^{+3.15} _{-3.14}	35.29 ^{+17.91} _{-14.64}	357.91 ^{+148.91} _{-112.66}	0.11 ^{+0.08} _{-0.08}	5.38 ^{+3.27} _{-3.66}
A.2	128.9 ^{+51.6} _{-17.1}	76.3	17.2 ^{+8.8} _{-6.2}	166.70 ^{+90.28} _{-77.77}	0.11 ^{+0.11} _{-0.07}	4.39 ^{+3.86} _{-3.28}	18.31 ^{+7.82} _{-6.59}	157.81 ^{+78.97} _{-69.06}	0.04 ^{+0.04} _{-0.02}	4.02 ^{+3.60} _{-2.76}
B.2	234.3 ^{+188.7} _{-30.8}	133.0	9.3 ^{+6.0} _{-2.6}	294.46 ^{+314.14} _{-157.81}	0.28 ^{+0.13} _{-0.17}	4.83 ^{+3.41} _{-2.89}	11.12 ^{+6.64} _{-3.48}	350.70 ^{+271.40} _{-194.18}	0.14 ^{+0.07} _{-0.07}	4.44 ^{+3.83} _{-2.76}
C.2	372.8 ^{+527.0} _{-58.2}	207.7	18.6 ^{+7.8} _{-7.4}	603.09 ^{+135.33} _{-263.11}	0.19 ^{+0.16} _{-0.13}	6.29 ^{+2.63} _{-3.50}	22.63 ^{+16.16} _{-7.29}	632.09 ^{+120.30} _{-206.55}	0.08 ^{+0.08} _{-0.06}	5.56 ^{+2.86} _{-3.05}

d) In lines 284-285, the authors state that the dust attenuations and metallicities of the bound star clusters agree with the galactic measurement of the host arc. However, comparing the values in Table 1 and the Cosmic Gems arc values in line 228, the star clusters' A_V values seem to be significantly higher than the host galaxy's, by about a factor 10. Is this expected for star clusters compared to their host galaxy?

>>>> Yes, as star clusters are localized stellar identities it is very common that at the location of the clusters we find different extinctions than at larger scales. A clear example can be seen for example in Sirressi et al 2021. There the star cluster photometry (in a similar way done for the cosmic gems arc clusters) and the spectroscopy of a kiloparsec scale region containing the clusters are analyzed via SED fit. The recovered average extinction for the whole knots containing 10s of clusters does not reflect that of the single clusters. More examples can be seen throughout the literature. Finally within the 68% confidence level the A_V of the clusters ranges from 0 to 0.4 mag. Also important to notice that extinction and age are linked, so the changes in extinctions are connected to the change in ages. The difference between galaxy-wide extinction value and star cluster values are not in disagreement. When fitting with SMC law, we retrieve values consistent with 0. We have added a sentence in Section 2.2 comparing A_V outputs.

e) In line 497, the redshift is fixed to 10.2 for the SED-modeling analysis. Given that this is a photometric redshift, with significant uncertainties of 0.2 and perhaps even larger systematic uncertainty, it would be useful to verify how the physical parameters of the star clusters depend on the redshift assumed for the fitting. This can be done by either leaving the redshift as a completely free parameter or by placing a Gaussian prior corresponding to the photometric redshift estimate of the host galaxy on it. The latter would in addition have the benefit to fold the redshift uncertainty into the physical parameter uncertainties.

>>>>> As the referee can evaluate from the available HST and JWST multiband photometry, we do not have a good detection of the star clusters below the JWST/F150W filter. The HST fluxes of the galaxy in the IR HST bands sample the Lyman break and therefore help to narrow down the redshift solution that is tightly constrained around the 10.2 (see delta function in the plot above). As we mention in the manuscript 0.2 error in redshift encloses the 95% of the solutions (2 sigma). In absence of reliable photometry across the Lyman break, the SEDs of the clusters have a shape that is insensitive to redshift variations and therefore requires a fixed redshift (even the gaussian distribution available in BAGPIPES would not work). However, during the analysis we tested the impact of the redshift uncertainty by assuming redshift 10 and 10.2 separately. We report in the table the recovered physical parameters of the clusters for these two assumptions. The changes in the recovered physical values is negligible. We notice that by mistake we have reported in Table 1 values that were recovered assuming $z=10$. We have updated the table (now Table 3 in Extended data) accordingly. The rest of the analysis presented in the draft was obtained using the solution for $z=10.2$.

Table 4 Comparisons between SED fit outputs estimated with BAGPIPES 1 Myr exp at $z = 10$ vs. $z = 10.2$. The reported masses are not corrected for magnification.

ID	BAGPIPES $z = 10$, exp 1 Myr, Calz				BAGPIPES $z = 10.2$, exp 1 Myr, Calz			
	Age [Myr]	$10^6 M_\star$ [M_\odot]	A_V [mag]	Z/Z_\odot [%]	Age [Myr]	$10^6 M_\star$ [M_\odot]	A_V [mag]	Z/Z_\odot [%]
A.1	$8.8^{+12.8}_{-3.0}$	$118.37^{+289.37}_{-71.78}$	$0.22^{+0.15}_{-0.15}$	$6.36^{+2.32}_{-3.08}$	$9.2^{+13.5}_{-3.2}$	$138.34^{+293.88}_{-87.99}$	$0.22^{+0.15}_{-0.14}$	$6.38^{+2.42}_{-3.08}$
B.1	$13.2^{+6.0}_{-3.3}$	$264.55^{+132.87}_{-106.78}$	$0.20^{+0.14}_{-0.10}$	$4.29^{+3.35}_{-3.27}$	$14.0^{+6.5}_{-4.2}$	$289.37^{+118.86}_{-137.65}$	$0.20^{+0.12}_{-0.12}$	$3.42^{+4.02}_{-2.49}$
C.1	$8.9^{+8.1}_{-2.7}$	$170.00^{+220.68}_{-94.27}$	$0.37^{+0.09}_{-0.16}$	$4.92^{+3.00}_{-2.98}$	$9.1^{+9.2}_{-2.8}$	$172.51^{+271.70}_{-100.24}$	$0.36^{+0.09}_{-0.16}$	$4.56^{+3.49}_{-2.88}$
D.1	$9.1^{+11.6}_{-3.5}$	$151.69^{+291.66}_{-98.63}$	$0.34^{+0.10}_{-0.17}$	$4.99^{+3.40}_{-2.86}$	$10.9^{+13.1}_{-3.9}$	$235.49^{+256.24}_{-154.93}$	$0.32^{+0.12}_{-0.16}$	$5.27^{+3.12}_{-3.03}$
E.1	$34.4^{+18.7}_{-13.8}$	$391.65^{+128.88}_{-125.37}$	$0.23^{+0.15}_{-0.16}$	$5.07^{+3.23}_{-3.32}$	$36.8^{+19.7}_{-16.4}$	$421.73^{+153.99}_{-151.94}$	$0.21^{+0.18}_{-0.15}$	$5.05^{+3.15}_{-3.14}$
A.2	$16.2^{+8.8}_{-6.0}$	$169.29^{+81.08}_{-86.04}$	$0.14^{+0.13}_{-0.09}$	$4.36^{+3.54}_{-2.99}$	$17.2^{+8.8}_{-6.2}$	$166.70^{+90.28}_{-77.77}$	$0.11^{+0.11}_{-0.07}$	$4.39^{+3.86}_{-3.28}$
B.2	$8.9^{+4.4}_{-2.9}$	$262.52^{+281.12}_{-154.16}$	$0.29^{+0.15}_{-0.15}$	$4.33^{+3.56}_{-2.61}$	$9.3^{+6.0}_{-2.6}$	$294.46^{+314.14}_{-157.81}$	$0.28^{+0.13}_{-0.17}$	$4.83^{+3.41}_{-2.89}$
C.2+D2	$19.2^{+7.4}_{-6.4}$	$619.24^{+124.77}_{-233.83}$	$0.19^{+0.16}_{-0.13}$	$5.91^{+2.72}_{-3.11}$	$18.6^{+7.8}_{-7.4}$	$603.09^{+135.33}_{-263.11}$	$0.19^{+0.16}_{-0.13}$	$6.29^{+2.63}_{-3.50}$

3. In line 470 the authors describe how the uncertainties on photometry and size were derived. Were the magnification uncertainties propagated into these quantities? If yes, did these include the lens modeling systematics?

>>>>>The photometry and size errors reported in the tables do not account for magnification uncertainties. We did not want to tie the observed measured quantities and derived SED fit parameters to 1 lensing model. In Table 1 we report the mass, r_{eff} , Σ_{star} (stellar density), and Π (dynamical age) by using 1 magnification value (the reference one). The errors associated with those physical quantities are estimated by using the same single

magnification value. The magnification error estimates are described in Section 1.3 of Methods. In Figure 6 of extended data, we evaluate the effect of magnification uncertainties on the derived physical parameters. The colored bands around the measured physical values show the uncertainties obtained in the analysis and reported in Table 1. These uncertainties are plotted against the change in magnification reported in the x axes. The green vertical bands show the range of variations in the measured physical parameters due to the magnification uncertainties. As we report in the main text, variations in the magnifications do not change the results: 1. Stellar densities remain well above the average observed for globular clusters (e.g., 10^3 Msun/pc^2), 2. Pi values remain significantly larger than 10, so these stellar systems are consistent with being gravitationally bound.

Minor comments:

4. In line 111: Perhaps also cite the recent work by Atek et al. 2023 here for the ionization power of young low-mass galaxies in the EoR (also again later, in line 382).

>>>Done

5. In line 180: For MACS0647-JD, there was a paper presenting the spectroscopic redshift measurement in addition to the photometric paper cited here (reference number 8), Hsiao et al. 2023b.

>>>Unfortunately we are already above the limit of 30 references.. We have omitted the second reference to Hsiao et al 2023b even if it is our own group work.

6. In Table 1: I am aware that the main text specifies that the stellar masses reported here are de-magnified, but I would nevertheless recommend also specifying this in the table or its caption, to make sure it is immediately visible and avoid possible confusion.

>>> Done

7. In line 399: Double 'the'.

>>>>done

8. In line 444, it is stated that the tangential magnifications were used for the radius measurements, the tables however report the total and the radial magnifications. While I understand the reason why this is done, I would nevertheless suggest to perhaps simply report the total and tangential magnifications in each table, to avoid confusion and be more clear to readers who are not necessarily experts on lensing.

>>>>>Done

Referee #2 (Remarks to the Author):

Please incorporate the points below into your comments to the Authors. For all other article types, such as review or progress articles, please provide comments to authors and editors in the boxes provided below.

Summary of the key results

The paper reports the discovery of young massive star clusters in a strongly lensed galaxy at redshift ~ 10.2 , when the Universe was only 420 Myr old.

The observations are from JWST/NIRCam and reveal five distinct clusters threaded along the arc, which drop out at F115W.

The clusters have a de-lensed size of $\sim 1 \text{ pc}$, making up 60% of total FUV light.

They are fairly young ($< 35 \text{ Myr}$), and are fairly massive for the era (10^6 Msun), resulting in extremely high surface densities, three orders of magnitude above local universe star clusters.

These are the earliest known proto-globular cluster candidates.

Originality and significance: if not novel, please include references.

If the redshift is confirmed spectroscopically, this will be the most magnified object at the highest redshift, resolved to the parsec scale. Based on the object's dropout in F115W, the photometric redshift is $z > 9$. Hence, this is a novel and significant observational result.

>>>> See reply to reviewer #1 regarding redshift determination.

Data & methodology: validity of approach, quality of data, quality of presentation.

Since most photometric and size measurement methodologies will be described in a paper in preparation, reviewing the analysis techniques has been challenging. I am requesting the latest draft of Bradley et al. in prep to gauge the data quality and analysis fully.

>>>> We have revised the text to make it clear to the reader that the methodology to measure photometry and size of the star clusters used in this work is published and tested. The only paper in preparation we cite, i.e. Bradley et al is the one related to the galaxy scale measurements. In that paper we will not discuss how star cluster sizes and fluxes are derived. We follow the guideline for Nature that says if a used method is published can be referenced in the text for reference and therefore, detailed descriptions can be omitted.

The observation is novel enough for this paper to merit publication. To increase the robustness of the reported physical parameters, I suggest the following improvements:

It is not trustworthy to measure sizes smaller than the PSF's HWHM. Many of the sizes quoted in Table 1 are below this limit. I am requesting a simulation demonstrating their technique's size and flux limit both in the image and source planes.

The sizes and, consequently, the fluxes are measured by modelling a point spread function or the shape of the clumps in extremely small cutouts (7x7 or 11x11 pix). Such a small cutout cannot fit the wings of the PSF, nor will it be able to estimate the background properly, separating it from the light of the arc. I am requesting a simulation demonstrating the effectiveness of their technique's total light recovery. The uncertainty in flux measurement from a small cutout will likely be very high, depending on the background level and will vary with wavelength.

>>>> The methodology used in the manuscript has been carefully tested with simulation in Messa et al 2019 (<https://ui.adsabs.harvard.edu/abs/2019MNRAS.487.4238M/abstract> see appendix) and Messa et al 2022 (<https://ui.adsabs.harvard.edu/abs/2022MNRAS.516.2420M> appendix). The software is designed to measure sizes of objects in the domain of "slightly" extended objects which can be modeled as simple analytic functions, the latter is created by convolving the PSF of the data with analytical profiles. In both Messa et al the software has been tested with simulations, by fitting synthetic sources and comparing derived fluxes also wrt aperture photometry fluxes. We stress that all the analysis presented in the manuscript has been done on box size of 11 px or larger centred on the sources. The 7x7 box size was used to attempt the photometry of the very faint E.2 and it is not included in the analysis. We have included detailed information about the fitting of the sources and tests we have performed during the analysis based on the box size at the end of this report. Regarding the simulations requested by the referee, we have performed them exploring the fluxes and sizes that overlap with the results shown in this manuscript. We have injected 400 simulated sources with intrinsic input size from 0.1px to 3 px (upper limit corresponds to the largest size measured in the cosmic gems clusters) and input fluxes between 26.5 and 29.5 ABmag in the F150W frame. The two plots show the recovered measurements. In the size plot (left) we see that our method does not recover intrinsic radii below 0.5 px (dotted horizontal line), while sizes at 0.6 px (as for cluster A.1) the differences are about 20% (dashed lines around plotted points). The agreement between input and output size is very good between 1 and 3 px. The right plot shows the input and output ABmag of the fitted sources (solid line is the one to one, dotted lines show 0.1 mag error). Overall the large majority of the sources have recovered fluxes within 0.1 mag down to magnitudes of 28.5 ABmag. Below these values uncertainties become larger. The faintest source in our sample is 28.3 ABmag (E.1) and has uncertainties of 0.2 mag that accounts for the 0.1 ABmag caused by the method.

Regarding the reliability of the R_{eff} measurements:

The simulations conducted in Messa et al 2019 and Messa et al 2022 show that the method (based on empirical PSFs) can recover intrinsic (PSF deconvolved) core radii (then transformed into R_{eff}) down to 0.4 px. The method developed in Messa et al 2019 is a python revisitation of the previously published software ISHAPE (Larsen 1999, <https://aas.aanda.org/articles/aas/pdf/1999/17/ds8625.pdf>), used to derive star cluster sizes in the literature throughout 2000-2015 period. Larsen 1999 also concluded from their tests that the software could recover intrinsic sizes down to 10 % of a pixel. Similar methods, where the fit of “slightly” extended objects in pixelised images is done by de-convolving the PSF shape from the fit of the data, are published in the literature. Vanzella et al 2019 (and reference therein) discuss the ability to recover sub-HWHM intrinsic sizes in lensed point-like sources. Also there, the authors report measurements of intrinsic R_{eff} (after deconvolution from PSF) to less than 1 px. We understand the concern of the reviewer and while simulations would suggest that these methods are able to derive intrinsic R_{eff} down to fraction of pixels and definitely smaller than HWHM, the estimates become increasingly less reliable as these limits are reached.

In the case of the cosmic Gems arc, we report the FWHM of the F150W to be 0.05” (measured on the empirical PSF reconstructed from stars in the final data) where the pixel scale is 0.02”. This corresponds to a HWHM = 1.25 pixel. The reported intrinsic measurements of R_{eff} (after PSF deconvolution) listed in Table 1 are all comparable to the HWHM of the data or larger, except for cluster A. While the method suggests we can achieve subpixel measurements of R_{eff} , we refer to cluster A as marginally resolved (line 225-226). It is important to notice that we provide R_{eff} measurements with a second method that is totally independent and based on forward modeling of intrinsic sources that from the source plane are projected into the image plane. The sizes derived in this way are in agreement with direct measurements done in the image plane. Therefore strengthening the conclusion that size measurements are reliable.

Regarding the reliability of flux measurements in addition to the simulations showed above we also provide a summary where we compare simple aperture photometry to the extracted photometry with our method, that we refer to as PSF-like photometry on slightly resolved systems in the attached PDF file at the end of this document. Our method works better in crowded regions where aperture photometry will be contaminated and difficult to disentangle, while our photometry is comparable to aperture photometry performances in isolated sources. We show the test in the reference filter, F150W, where the shape of the source is fitted and fixed, thus if any systematic is introduced and will be propagated to all the other bands.

We also address the last concern, related to the size of the box used to fit the source shape and disentangle the local background (which includes diffuse light of the galaxy). We fit simultaneously, shape, flux and local background. The

simulations above suggest that we are indeed able to disentangle these three parameters. We include the cutouts produced by the software and that we use to check that the fit is properly working. We have tested several box sizes. In the PDF file, we show the recovered values of fluxes and intrinsic sizes (after de-convolution from the PSF) recovered using different box sizes. We notice that a box size of 11x11 px is about 3.2 times the FWHM. The large uncertainties associated to D.1 and E.1 are in line with the faintness of these sources and the trends shown in the simulation described above.

Section 2.1 of Methods has been updated to reflect these tests and considerations.

Even though three sets of SED fitting have been performed, the assumptions on SFH are not varied enough to probe the range of possible ages. The exponential mode assumes a very short $\tau=1$ Myr, essentially equivalent to the burst model. This is reflected in the similar physical properties recovered from these two fits. I suggest allowing for multiple bursts and longer taus to probe possibilities of older age.

In answering reviewer #1 we tested different assumptions (in particular extinction and redshift and different software, prospector). We include here below a table comparing physical parameters recovered with the reference fit vs. the ones recovered letting tau be a free parameter. The recovered mass-weighted ages are older but still consistent within uncertainties with those recovered with a fixed tau. The recovered tau, in the latter fit does not change significantly. From this experiment, we conclude that changing assumptions in terms of adopted extinction, redshift variations, broader SFH, do not change the conclusions of the manuscript, i.e. that we are detecting dense and bound star clusters at redshift 10.

Table 6 Comparisons between SED fit outputs estimated with BAGPIPES. Both fit has $z = 10.2$, & Calzetti attenuation law but differ on the SFH assumptions. In the first run we assume a very short $\tau = 1$ Myr while in the second run we let it free to vary. The reported ages in the latter case are mass-weighted. The reported masses are not corrected for magnification.

ID	Magnification models		BAGPIPES $\tau=1$ Myr, $z = 10.2$, & Calzetti					BAGPIPES τ =free, $z = 10.2$, & Calzetti			
	$\mu_{\text{lenstool-B}} (\mu_r)$	$\mu_{\text{glafic}} (\mu_r)$	Age [Myr]	$10^6 M_*$ [M_\odot]	A_V [mag]	Z/Z_\odot [%]	τ [Myr]	Age [Myr]	$10^6 M_*$ [M_\odot]	A_V [mag]	Z/Z_\odot [%]
A.1	$122.0^{+48.8}_{-17.8}$	76.9	$9.2^{+13.5}_{-3.2}$	$138.34^{+293.88}_{-87.99}$	$0.22^{+0.15}_{-0.14}$	$6.38^{+2.42}_{-3.08}$	60.79	$27.90^{+21.74}_{-14.82}$	$192.02^{+135.05}_{-147.57}$	$0.08^{+0.08}_{-0.05}$	$6.39^{+2.40}_{-3.13}$
B.1	$212.6^{+152.8}_{-25.4}$	153.8	$14.0^{+6.5}_{-4.2}$	$289.37^{+118.86}_{-137.65}$	$0.20^{+0.12}_{-0.12}$	$3.42^{+4.02}_{-2.49}$	65.22	$19.15^{+14.53}_{-7.31}$	$162.58^{+97.53}_{-5.19}$	$0.04^{+0.06}_{-0.03}$	$2.16^{+2.54}_{-1.55}$
C.1	$280.0^{+270.7}_{-27.2}$	200.9	$9.1^{+9.2}_{-2.8}$	$172.51^{+271.70}_{-100.24}$	$0.36^{+0.09}_{-0.16}$	$4.56^{+3.49}_{-2.88}$	60.52	$15.68^{+16.10}_{-9.40}$	$171.97^{+137.98}_{-69.69}$	$0.20^{+0.12}_{-0.11}$	$3.84^{+3.36}_{-2.33}$
D.1	$349.5^{+474.6}_{-56.5}$	288.8	$10.9^{+13.1}_{-3.9}$	$235.49^{+184.93}_{-135.99}$	$0.32^{+0.12}_{-0.16}$	$5.27^{+3.12}_{-3.03}$	69.27	$18.39^{+22.90}_{-10.53}$	$167.89^{+176.63}_{-86.57}$	$0.19^{+0.13}_{-0.11}$	$5.08^{+2.97}_{-3.12}$
E.1	$527.3^{+708.6}_{-30.1}$	516.4	$36.8^{+16.7}_{-16.4}$	$421.73^{+154.93}_{-151.94}$	$0.21^{+0.18}_{-0.15}$	$5.05^{+3.15}_{-3.14}$	67.05	$54.85^{+42.58}_{-30.97}$	$314.15^{+176.71}_{-68.91}$	$0.19^{+0.15}_{-0.13}$	$4.15^{+3.75}_{-2.70}$
A.2	$128.9^{+51.6}_{-17.1}$	76.3	$17.2^{+8.8}_{-6.2}$	$166.70^{+90.28}_{-77.77}$	$0.11^{+0.11}_{-0.07}$	$4.39^{+3.86}_{-3.28}$	65.64	$27.56^{+22.23}_{-13.42}$	$123.81^{+97.75}_{-35.06}$	$0.05^{+0.07}_{-0.03}$	$3.63^{+3.32}_{-2.34}$
B.2	$234.3^{+188.7}_{-30.8}$	133.0	$9.3^{+6.0}_{-2.6}$	$294.46^{+314.14}_{-137.81}$	$0.28^{+0.13}_{-0.17}$	$4.83^{+3.41}_{-2.89}$	64.48	$15.87^{+15.57}_{-7.76}$	$270.99^{+209.38}_{-114.48}$	$0.16^{+0.12}_{-0.10}$	$4.43^{+3.55}_{-2.59}$
C.2	$372.8^{+527.0}_{-58.2}$	207.7	$18.6^{+7.8}_{-7.4}$	$603.09^{+355.33}_{-263.11}$	$0.19^{+0.16}_{-0.13}$	$6.29^{+2.63}_{-3.50}$	69.46	$34.25^{+16.40}_{-19.41}$	$474.27^{+173.05}_{-48.73}$	$0.13^{+0.15}_{-0.09}$	$3.79^{+3.77}_{-2.60}$

The SED fitting has been performed with fixed assumptions on ionizing parameters, narrow SFH, IMF, etc. While the current photometry will not be able to reveal any new information on these parameters, changing the fixed value will give very different stellar population properties. I am interested in how these clusters' age and stellar mass change due to different IMF-alpha (indications of top-heavy IMF at these redshifts), ionization parameters, and broader SFH. Have the authors tried fitting with other SED methods such as Prospector, Cigale, and Dense Basis? How do the stellar population properties change when non-parametric SFH are assumed?

>>>> We stress here that we let the ionisation parameter (U) change in the fit, but we do not recover significant constraints (no convergence in the posterior distributions). Therefore they have been omitted in the manuscript.

We agree with the assessment of the referee. Letting the parameter grid fully free on SFH, metallicity, IMF-alpha slope, will provide a similar good fit to the cluster light. However that result will not truly constraint the intrinsic physical properties of the clusters. We are studying the light produced in a compact stellar region of about 1-2 pc in size with a 2D shape that is compatible with being a spherical-like system. We have to use all the information we have at hand to constraint the physical properties of these objects. Star clusters in the local universe (young and old, massive or not) do not show presence of continuous star formation, even when we fit their stellar continuum and have

absorption line features that are sensitive to SFH. A significant example is the analysis of one of the most massive star clusters reachable in the local universe <https://ui.adsabs.harvard.edu/abs/2016MNRAS.457..809C/abstract> formed during a merger event. No trace of continuum star formation or multiple burst events can be found in the combined photometry+spectroscopy. Our choice of fixing the SFH to a single burst is indeed guided from physical information we have at hand. We have access to only 6 datapoints covering from 1000 to 4000 AA rest frame. This range is sensitive to the presence of young (massive) stars. The choice of using BPASS with a higher stellar mass cutoff was motivated for the search of extremely blue stars that standard models do not account for. However, even these models provide similar results. We point out that the fit to the galaxy light has been done without these constraints of SFH and the results we get by analyzing the star clusters is in agreement with the ages we constrain.

The lack of spectral line features and access to only broadband photometry does not provide any leverage to test IMF variations. We can report IMF variations but our data do not have enough information to really disentangle this parameter, therefore we prefer to remain conservative. We have a NIRSpec/MIRI Cycle3 spectroscopy program accepted. We will definitely explore this question with more data at hand. Finally as replied to the question pointed out by reviewer #1, we have fitted the cluster light using prospector. In that case we have assumed a SSP, a standard assumption for star cluster analyses in the local universe.

We have revised Section 2.2 of Methods to reflect the points raised on SED fitting degeneracies by both reviewers. We have discussed the different output parameters and checked whether they would affect the conclusions of finding dense and bound star clusters.

Appropriate use of statistics and treatment of uncertainties.

While it has been challenging to get a full picture of the analysis since many of the details will be presented in a paper in preparation, I believe many of the uncertainties on derived stellar properties will increase when they consider the suggestions presented in the previous section.

We have addressed all the points raised by reviewer #2. There is no paper in preparation. The method used to analyze the data is published and has been used throughout the literature in the following publications:

Messa, Matteo, Dessauges-Zavadsky, Miroslava, Adamo, Angela, Richard, Johan, and Claeysens, Adélaïde, "Properties of the brightest young stellar clumps in extremely lensed galaxies at redshifts 4 to 5", MNRAS accepted, <https://ui.adsabs.harvard.edu/abs/2024arXiv240214920M>

Claeysens, Adélaïde, Adamo, Angela, Richard, Johan, Mahler, Guillaume, Messa, Matteo, and Dessauges-Zavadsky, Miroslava, "Star formation at the smallest scales: a JWST study of the clump populations in SMACS0723", MNRAS 2023, <https://ui.adsabs.harvard.edu/abs/2023MNRAS.520.2180C>

Messa, Matteo, Dessauges-Zavadsky, Miroslava, Richard, Johan, Adamo, Angela, Nagy, David, Combes, Françoise, Mayer, Lucio, and Ebeling, Harald, "Multiply lensed star forming clumps in the A521-sys1 galaxy at redshift 1", MNRAS 2022, <https://ui.adsabs.harvard.edu/abs/2022MNRAS.516.2420M>

Messa, Matteo, Adamo, Angela, Östlin, Göran, Melinder, Jens, Hayes, Matthew, Bridge, Johanna S., and Cannon, John, "Star-forming clumps in the Lyman Alpha Reference Sample of galaxies - I. Photometric analysis and clumpiness", MNRAS 2019, <https://ui.adsabs.harvard.edu/abs/2019MNRAS.487.4238M>

Vanzella, Eros, Claeysens, Adélaïde, Welch, Brian, Adamo, Angela, Coe, Dan, Diego, Jose M., Mahler, Guillaume, Khullar, Gourav, Kokorev, Vasily, Oguri, Masamune, Ravindranath, Swara, Furtak, Lukas J., Hsiao, Tiger Yu-Yang, Abdurro'uf, Mandelker, Nir, Brammer, Gabriel, Bradley, Larry D., Bradač, Maruša, Conselice, Christopher J., Dayal, Pratika, Nonino, Mario, Andrade-Santos, Felipe, Windhorst, Rogier A., Pirzkal, Nor, Sharon, Keren, de Mink, S. E., Fujimoto, Seiji, Zitrin, Adi, Eldridge, Jan J., and Norman, Colin, "JWST/NIRCam Probes Young Star Clusters in the Reionization Era Sunrise Arc", ApJ 2023, <https://ui.adsabs.harvard.edu/abs/2023ApJ...945...53V>

The uncertainties on the derived physical properties do not impact the results. We have compiled a table with all different SED fits, including Prospector assuming a SSPs. We see that overall ages, masses, extinction changes within the associated 68% confidence level even in the case of Prospector. We have added in Figure 2 lines showing stellar densities for equal masses. For example in the fit done with Prospector the resulting intrinsic mass of cluster A would be $4 \times 10^5 M_{\text{sun}}$. Resulting in slightly lower stellar densities that are currently quoted in table 1 and Figure 2, but still within the error bars (see lines 586-590). Therefore, we conclude that the results remain consistent. We thank the referee for asking these tests that have hopefully removed any doubt.

Conclusions: robustness, validity, reliability

The observation is reliable, but the photometry and SED fitting techniques are not robust (see section 3).

>>>>We have addressed the robustness of the photometry and performed several test in attached PDF

Suggested improvements: experiments, data for possible revision

Please see section 3 for suggested improvements.

>>>>We have addressed all concerns

References: appropriate credit to previous work?

The references seem adequate.

Clarity and context: lucidity of abstract/summary, appropriateness of abstract, introduction and conclusions.

The paper seems hastily written and is missing a discussion on the implications of finding these star clusters at $z > 10$ on globular cluster formation and the assembly of galaxies. The observation is extremely exciting, but the paper reads very dryly and will not be of interest to someone who is not in this field.

>>>> The manuscript has undergone significant changes and addresses this point. We have rewritten the discussion including relevance of this discovery for:

1. Black hole seed formation
2. Enabling formations of very massive stars via runaway stellar mergers as well as host massive stars. Both channels can explain the metal enrichment found in some reionization era galaxies
3. We have added comparisons with simulations that can explain the formation of such dense stellar objects, predict their formation based on the relevant physical properties of their host galaxy, and predictions related to the launch of outflows that favor the escape of ionizing radiation.
4. We have compared this finding with prediction of GC formation in MW progenitors.
5. We have also included the intrinsic FUV magnitude of the Cosmic Gems arc. This galaxy overlaps well with the faint detected galaxies at this redshift suggesting that star cluster formation could be an important mode for reionization galaxies.

Cluster A.1

F150W(FWHM=0.05"). Box size 11px = 0.22"

Flux=3.18 nJy
R_eff, obs = 0.6 px

F150W(FWHM=0.05"). Box size 9px = 0.18"

Flux=3.12 nJy
R_eff, obs = 0.6 px

F150W(FWHM=0.05"). Box size 13px = 0.26"

Flux=3.16 nJy
R_eff, obs = 0.6 px

source cutout

Best model

residuals

Effect of size box on recovered physical parameters. In each cutout the black dots is the initial position, the red shows the best fitted one. The bar shows nJy. Please notice that the bar in the residual plots shows values that are about a factor of 10 smaller. The change of box size for cluster A has no effect on the recovered fluxes and sizes.

Cluster B.1, C.1, D.1, E.1

F150W(FWHM=0.05"). Box size 15px = 0.3"

$F(b1)=4.03$ nJy, $R_{\text{eff, obs}}(b1) = 1.1$ px
 $F(c1)=2.74$ nJy, $R_{\text{eff, obs}}(c1) = 0.4$ px
 $F(d1)=2.67$ nJy, $R_{\text{eff, obs}}(d1) = 1.2$ px
 $F(e1)=1.76$ nJy, $R_{\text{eff, obs}}(e1) = 1.5$ px

F150W(FWHM=0.05"). Box size 17px = 0.34"

$F(b1)=3.97$ nJy, $R_{\text{eff, obs}}(b1) = 0.8$ px
 $F(c1)=2.9$ nJy, $R_{\text{eff, obs}}(c1) = 0.5$ px
 $F(d1)=1.8$ nJy, $R_{\text{eff, obs}}(d1) = 0.3$ px
 $F(e1)=2.5$ nJy, $R_{\text{eff, obs}}(e1) = 2.85$ px

F150W(FWHM=0.05"). Box size 19px = 0.38"

$F(b1)=4.02$ nJy, $R_{\text{eff, obs}}(b1) = 1.2$ px
 $F(c1)=2.74$ nJy, $R_{\text{eff, obs}}(c1) = 0.2$ px
 $F(d1)=1.58$ nJy, $R_{\text{eff, obs}}(d1) = 0.1$ px
 $F(e1)=3.27$ nJy, $R_{\text{eff, obs}}(e1) = 4$ px

source cutout Best model residuals

Effect of size box on recovered physical parameters in case of complex source configurations. In each cutout the black dots is the initial position, the red shows the best fitted one. The bar shows nJy. Please notice that the bar in the residual plots shows values that are about a factor of 10 smaller. The change of box size does not have significant impact for source B.1 and C.1, these are the brightest sources in the complex. This is not the case for D.1 and E.1, the faintest sources in this region, where the solutions remain degenerate. For increasingly larger box sizes, the fit prefers solutions with E.1 fitted with increasingly larger ellipticity gaussians, while D.1 becomes a PSF like system. Significant flux is added to E.1 from the diffuse light of the arc for larger box sizes. We decided to use the most conservative estimates produced by the box size 15 px, where E is the faintest source in the complex, in line with the visual appearance of the data. With this choice of box size all the sources have an ellipticity of close to 2, which is also in line with the slightly elongated 2D appearance. We notice that large uncertainties in the size and flux determinations of D.1 and E.1 are reflected in the estimated uncertainties, which are large for these two clusters (see Table1 and Table 2 in manuscript). Finally, the solution obtained with box size 15x15 px is recovered for all the sources, including D.1 and E.1, if we fix the ellipticity of these latter sources when fitting box sizes 17 and 19.

Cluster E.2

F150W(FWHM=0.05") - reference filter. Box size 7 px = 0.14"

source cutout

Best model

residuals

E.2 is among the faintest sources in the arc. A smaller box size has been used to minimise the effect of contaminations from other bright clusters in the proximity. However, even with a small box the fit of the source fails because of its faintness. The residuals are comparable to the initial fluxes within a factor of 2. **This source is not included in the analysis, it is the only one where we used a small box size.** It also shows the limitations of the method that underperform for sources detected below 5sigma.

Photometry test. We compare a simple flux count within a given aperture (we do not account for aperture correction and we do not subtract for local sky bkg) versus the final magnitude determined with our method (PSF-like fit). We see very good agreement in the recovered photometry when an aperture of 4 pixels (3.2 times the HWHM) which can be applied to cluster A.1, A.2, B.2, C.2 with minimal contamination. For the clusters B.1, C.1, D.1, E.1 we use a smaller radius of 2.5 px (2 times the HWHM). For these clusters we see the following $\Delta_{\text{mag150}} (\text{ap}_{\text{phot}} - \text{PSF}_{\text{phot}}) = (0.46629498, 0.12666479, 0.45112366, 0.67426538)$ for B.1, C.1, D.1, E.1 respectively. These differences are not surprising as for smaller aperture the aperture correction will be significantly larger and the contamination from the local emission more difficult to disentangle due to the overlap of several PSF-like source. Overall, the PSF fit appears to be a robust method to estimate photometry for more isolated point-like source and a more reliable approach in crowded regions (in the same way DOLPHOT operates in resolved stellar fields).

Reviewer Reports on the First Revision:

Referees' comments:

Referee #1 (Remarks to the Author):

After reading the revised version of the manuscript and viewing the extra figures and material provided, I find all of my comments regarding the SED-fitting procedure answered and resolved in a satisfactory manner. The manuscript reads very well and presents compelling results. I am therefore happy to recommend it for publication in general.

However, I am still concerned about the presentation of the redshift measurement: The discussion and figures provided by the authors in response to my report indeed convince me that the photometric redshift estimate is fairly robust. The authors also explain why the spectroscopic measurement is not included (non-detection of targeted emission lines), which I appreciate, and that the low-redshift solution would present prominent emission lines in the spectra, which are however also not detected. Nevertheless, neither the JWST dropout figure, nor the discussion included in the answer to my report, which are a compelling support to the high-redshift solution, have been added to the paper. I strongly recommend these, or something similar, to be included in the paper in support of the high-redshift solution. Right now, the manuscript still reads as the first version concerning the redshift measurement, i.e. assuming the object lies at $z=10$ and deferring the discussion to future work. I think the paper would greatly benefit from a discussion of the photometric redshift solution.

Referee 2:

Manuscript#: 2024-01-00352A

Corresponding Author: Angela Adamo

Title: JWST reveals bound star clusters 460 Myr after the Big Bang

I thank the authors for responding to my comments, revising the manuscript, and providing the suggested analysis and plots. I have revisited the manuscript with keen interest and maintain my enthusiasm for the subject matter. I am pleased with the inclusion of plots demonstrating the robustness of their photometric technique, which was a primary concern in the previous review. However, upon revisiting the question regarding redshift determination raised by Referee#1 and examining the provided images, I have developed concerns. I strongly recommend that the authors elucidate in this manuscript the methodology behind determining the redshift instead of deferring the analysis to a forthcoming paper. I am requesting that the authors:

1. Either include a dedicated section in this paper detailing their approach to redshift determination from photometry.
2. Alternatively, consider resubmitting this paper after the publication of Bradley et al. (in prep).

Clarifying the redshift determination process in the current manuscript would greatly enhance the understanding and credibility of the findings.

Data & methodology: validity of approach, quality of data, quality of presentation

1. The points raised by Referee # 1 regarding the redshift determination made me go back and re-check the analysis, now with the new information that this object is undetected in the available (albeit non-optimal) spectroscopy. I can see a faint hint of something in F125W (see below). This is also reflected in the photometric fit figure, a flux of 0.1 ± 0.03 (μJy). Given that the photometric redshift solution of this object hinges on the F115W drop-off nature of this object, the F125W carries crucial information for the fit. There is no mention of F125W in the manuscript. In the fit provided, the F125W flux is 2-sigma above the $z=10.2$ solution but within 1-sigma of the $z=2.5$ solution. Hence, the F125W detection does not favour the $z=10.2$ solution.
2. The manuscript lacks details regarding the upper limits on flux in the non-detection bands. The selection of bluer filters employed in the fits, along with their upper limits or actual fluxes

if detected, could significantly impact the posterior distributions of the redshifts. Therefore, it is imperative to provide this information for a more comprehensive understanding of the redshift determination process.

Figure A: Fits provided by the authors in referee response. Please clarify the $P(z)$ plot provided, which has an arbitrary y-axis. Where is the $z \sim 2$ solution?

3. Was the above fit conducted for the combined photometry of the system? I performed some quick fits using eazy-py on the individual clump photometry provided in Table 2. Depending

on the upper limit choice for the blue bands, I obtained redshift solutions ranging from 1.5 to 10.5. This variability persists when I sum up the fluxes and fit on the integrated photometry. I was unable to replicate the sharp $z=10.2$ solution depicted in Figure A above. Instead, I encountered numerous dusty solutions at $z < 2$ and extreme emission line solutions at various redshifts. It is essential to report the flux and errors in all filters used in these fits for reproducibility purposes. Additionally, please elucidate how the upper limits were handled in the fit.

4. Please include the full version of the above figure (with corrected $P(z)$ plot) in the paper, including image cutouts in all HST+JWST filters.

1. “The fit to the combined HST and NIRCcam photometry provides as best redshift estimate $z \sim 10.2_{+0.2}$ (95% confidence -0.2 level) and excludes lower redshift solutions (Bradley et al. in prep.).

The Method Section jumps to SED fitting and lens modelling, both of which are fixing $z=10.2$, without first showing how this redshift is derived. Please add a section in Method on redshift determination.

2. Please report in the paper that spectroscopy is available but the source is undetected (with the same explanation that has been provided in the referee response).

3. “The lower redshift solution is a poor fit to the observed data and would be discarded by spectroscopy at hand as it should have optical strong line emissions.”

In the referee response, please include a simulated spectra from the best fit SED (both $z=2.5$ and $z=10.2$ solutions) of the photometry, at the resolution of your available spectroscopic observation. Please show the available 2D spectra and the extraction.

1. Appropriate use of statistics and treatment of uncertainties.

a. In the paper, it is noted that the magnifications have been calculated with $z=10.2$. It is not clear whether the uncertainty in the redshift (± 0.2) is included in the reported magnification uncertainty. Will the object have such high μ if $z=10$ or 10.4 ? Has this uncertainty been propagated to the demagnified physical properties calculation?

- b. In the paper, it is noted that the SED fitting have been performed with $z=10.2$. How has the error in the redshift been propagated?
- c. The AB magnitude errors on the fluxes reported in Table 2 are symmetric. The error bars on F_{ν} in Figure A are also symmetric. Please clarify which one is correct.

4. Conclusions: robustness, validity, reliability.

The major conclusion of this paper is dependent on the redshift determination. Sufficient information on the photometric redshift fitting has not been provided in the paper (no section in Methods, no figures, no $P(z)$ plot). This undermines the robustness of this analysis.

5. Clarity and context: lucidity of abstract/summary, appropriateness of abstract, introduction and conclusions

Please consider removing “460 Myr” from the title. Please report on the abstract that the redshift is photometric.

Author Rebuttals to First Revision:

Referees' comments:

Referee #1 (Remarks to the Author):

After reading the revised version of the manuscript and viewing the extra figures and material provided, I find all of my comments regarding the SED-fitting procedure answered and resolved in a satisfactory manner. The manuscript reads very well and presents compelling results. I am therefore happy to recommend it for publication in general.

However, I am still concerned about the presentation of the redshift measurement: The discussion and figures provided by the authors in response to my report indeed convince me that the photometric redshift estimate is fairly robust. The authors also explain why the spectroscopic measurement is not included (non-detection of targeted emission lines), which I appreciate, and that the low-redshift solution would present prominent emission lines in the spectra, which are however also not detected. Nevertheless, neither the JWST dropout figure, nor the discussion included in the answer to my report, which are a compelling support to the high-redshift solution, have been added to the paper. I strongly recommend these, or something similar, to be included in the paper in support of the high-redshift solution. Right now, the manuscript still reads as the first version concerning the redshift measurement, i.e. assuming the object lies at $z=10$ and deferring the discussion to future work. I think the paper would greatly benefit from a discussion of the photometric redshift solution.

The draft focusing on redshift estimates and analysis of the galaxy property, ancillary data and lensing model is now submitted to ApJ and accessible in the arXiv.

<https://ui.adsabs.harvard.edu/abs/2024arXiv240410770B/abstract>

We have added the reference to the manuscript in our resubmission and omit duplication of plots and analyses in this draft.

Referee #2 (extracted from pdf file):

I thank the authors for responding to my comments, revising the manuscript, and providing the suggested analysis and plots. I have revisited the manuscript with keen interest and maintain my enthusiasm for the subject matter. I am pleased with the inclusion of plots demonstrating the robustness of their photometric technique, which was a primary concern in the previous review. However, upon revisiting the question regarding redshift determination raised by Referee#1 and examining the provided images, I have developed concerns. I strongly recommend that the authors elucidate in this manuscript the methodology behind determining the redshift instead of deferring the analysis to a forthcoming paper. I am requesting that the authors:

1. Either include a dedicated section in this paper detailing their approach to redshift determination from photometry.
2. Alternatively, consider resubmitting this paper after the publication of Bradley et al. (in prep).

Clarifying the redshift determination process in the current manuscript would greatly enhance the understanding and credibility of the findings.

We understand the referee's concerns. We have now submitted and shared with the community the manuscript fully dedicated to redshift estimate, galaxy physical properties, lensing model.

<https://ui.adsabs.harvard.edu/abs/2024arXiv240410770B/abstract>

All the questions raised by review 2 are addressed there. We notice that fluxes for the host galaxies are listed in Bradley et al 2024. The referee has extrapolated measurements from plots which definitely lead to propagation of uncertainties. In Bradley et al 2024 we derive redshift estimates with different codes and approaches like by fitting the SEDs of the entire arc or segments of the arc, all converging to redshift 10.2. We list in the Bradley et al 2024 the measured fluxes of the galaxy, the redshift estimates performed with two different softwares, the lack of detection in spectroscopy and the upcoming cycle3 NIRSPEC observations targeting the Lyman break. We prefer to not duplicate information, analysis, plots presented in the connected ApJ publication. Therefore we have not extended in this manuscript the redshift estimation.

We address here other comments from referee #2:

a. In the paper, it is noted that the magnifications have been calculated with $z=10.2$. It is not clear whether the uncertainty in the redshift (± 0.2) is included in the reported magnification uncertainty. Will the object have such high μ if $z=10$ or 10.4 ? Has this uncertainty been propagated to the demagnified physical properties calculation?

We find that the small change in redshift results in only a few per cent variations on the derived magnification values, while uncertainties related to the lensing model and included in Table 2 of the Extended Data section are significantly larger. Thus the uncertainties on the redshift does not change any of the results reported in our analysis.

b. In the paper, it is noted that the SED fitting has been performed with $z=10.2$. How has the error in the redshift been propagated?

We have tested how changing the redshift to values between 0.2 changes the recovered values. This was included in the answers to the previous report. We see no significant changes in recovered physical properties within the 68% uncertainties.

c. The AB magnitude errors on the fluxes reported in Table 2 are symmetric. The error bars on F_{ν} in Figure A are also symmetric. Please clarify which one is correct.

We derive flux_errors and convert them into magnitude errors using the standard relation $\text{mag_err} = 1.086 * \text{flux_err} / \text{flux} = 2.5 / \ln(10) * \text{flux_err} / \text{flux}$, which is derived from the general formula for error propagation under the assumption of $\text{flux_err} \ll \text{flux}$. So the error bars should appear symmetric both in flux and magnitude. Please notice that the SED figure shows linear y-axes.

Editor Comments

Dear Professor Adamo

Your revised manuscript, "JWST reveals bound star clusters 460 Myr after the Big Bang", has been seen again by the original referees, whose comments are attached below. While they find your work of interest, they have raised points that need to be addressed before we can make a decision on publication. We will need to consider your response to these concerns in the form of a revised manuscript accompanied by a list to explain your revisions. You will also need to make some editorial changes to your paper so that it is as brief as possible and complies with our Guide to Authors (<https://www.nature.com/nature/for-authors>).

The referees still have considerable reservations about the redshift determination. Referee 2 recommends a full explanation of the procedure, as he was unable to reproduce your redshift. Referee 1 recommends including the dropout figure and associated discussion in the manuscript. You have plenty of room in the Methods section and are under the limit for extended data figures.

I would like to make clear, however, that this is your final chance to convince the referees. Given the flood of papers into Nature, we cannot go through multiple rounds of review during which some creeping convergence takes place. If the next version is not completely convincing to the referees, we will be forced to terminate the process.

The companion paper where the host galaxy is presented and redshift determinations are derived has been submitted to the journal and available on arXiv. We prefer to not duplicate information in this manuscript, which focuses on the star clusters. This has been stressed on our answer to both referee comments.

Please provide the manuscript fully double spaced and in 12pt font. I very strongly recommend against using the SN template, which has not been remotely optimized for Nature.

The layout fully follows the expected Nature submission format, except in the references which are currently listed at the end of Methods. We have not managed to solve this problem. Rather than delay submission we can iterate with the editor office during the publication phase to ensure that it can be adapted.

Please note that the presentation of a Nature paper has undergone significant changes in recent years to improve its readability and navigability online. Specifically, in most instances any supplementary text and data figures/tables can now be integrated into the main paper rather than presented in a separate Supplementary Information file. An overview of the key features and differences may be found in the Composition of a Nature Paper (http://s3-service-broker-live-19ea8b98-4d41-4cb4-be4c-d68f4963b7dd.s3.amazonaws.com/uploads/ckeditor/attachments/7824/3g_Paper_composition.pdf).

This journal strongly supports public availability of data and custom code. Please place the data and code used in your paper into a public data repository, or alternatively, present the data as Supplementary

Information. If data and code can only be shared on request, please explain why in your Data Availability Statement, and also in the correspondence with your editor. Please note that for some data types, deposition in a public repository is mandatory - more information on our data deposition policies and available repositories appears below.

The Data availability section is at the end of Methods. Hopefully it is better visible in the current format.

LENGTH: We estimate the current length of your paper to be ~1950 words, which exceeds our usual limit by a considerable margin. With three display items as at present, the main text of the revised version should be no more than ~1800 words. Keep in mind that important technical details that are not central to the main message of the paper can be moved into the Methods section or, if necessary, a Supplementary Information section (see below).

The length of the main text is now ~1800 words.

TITLES: Titles cannot exceed 75 characters (including spaces); they must not contain punctuation. We prefer to avoid active verbs "reveals", and I think the time is past where it is necessary to name JWST in the title.

The title has been updated

SUMMARY PARAGRAPH: All Nature papers begin with a fully referenced paragraph, typically no longer than 200 words. The present background is very vague. Please be ****clear and specific**** about what has recently be claimed about high-z galaxies based upon JWST observations. The paragraph continues with a 1-sentence statement of the main observational result starting with 'Here we report' or an equivalent phrase. Please do not claim "discovery". Instead, you should phrase it as "Here we report observations of ...". The idea is to cleanly separate what you **saw** from what you inferred. It concludes with 2 to 3 sentences putting the main findings into the context established so it is clear how the results described in the paper have moved the field forward. ****In some cases it may be necessary to exceed this limit in order to explain complex material for readers in other fields – in such cases, summary paragraphs can be up to 230 words in length. The extra length, however, is for introduction and context, and not for additional technical information.**** Please omit the final sentence of the paragraph. The point will be obvious to experts and meaningless to everyone else, so it conveys no useful information.

The summary has been updated accordingly. It contains 194 words. The final sentence of the abstract has been changed and it now links back to the opening two sentences that set the stage. This should clarify how this result advances the field. We would like to note that the simple detection of star clusters in such a faint very high-redshift galaxy is a leap forward in our understanding of how star formation proceeds in early galaxies. That is why in the initial submission we tried to stress the aspect of this being a discovery that has implications to many different aspects of galaxy assembly and evolution. We understand that claim a discovery is not in the journal policy and we respect that.

MAIN TEXT: Further introductory material in the main text of the paper should not be necessary.

We have shortened the introduction to the bare minimum to explain where the Cosmic Gems Arc is located and what are the host galaxy physical properties which are relevant to then compare with the star clusters..

The PI name and program number (line 149) belong in the Data Availability section (which is currently missing).

This has been moved to the Data Availability section at the end of Methods. The Data Availability Section is after Methods.

Any discussion at the end of the paper should also be brief, and not repeat what is already written in the initial summary paragraph.

We have reorganized and reduced the main text. The conclusive discussion focuses on the implication that such high stellar densities can have for the stellar population residing in star clusters (and thus link to multiple stellar pop seen in GC at redshift zero). We conclude with a short paragraph to mention what will be the fate of these star clusters. We hope the manuscript reads more as a magazine article. We have strived to put into context each result as it is presented and not follow the classic division results, discussion, conclusions.

STATISTICS: Authors should ensure that any statistical analysis used is sound and that it conforms to the journal's guidelines (see <https://www.nature.com/nature/for-authors/formatting-guide> for guidance).

We have checked all statistics and errors. We have addressed the referee #2 concern on the symmetry of the plotted error bars.

METHODS: At the end of the main text document (after the main figure legends), there should be a section entitled "Methods", which provides a more detailed discussion of the additional methodological information that would allow other researchers to replicate the results (we define "Methods" quite broadly, so this is not limited to details of experimental protocols – supplementary discussion and analysis can also be included). The Methods section will not appear in the print version but will be fully copy-edited and appear online in the full-text HTML and PDF versions. The Methods section should be written as concisely as possible but should contain all elements necessary to allow interpretation and reproduction of the results. If there are additional references in the Methods section, their numbering should continue from the last reference in the main paper, and the list should follow the Methods section. If the methods require chemical structures, figures or tables, these should be supplied as Extended Data (see below). For mathematically complex methods, or methods that require an unusually large number of figures or tables (beyond what can be accommodated as Extended Data), the entire Methods section should instead be supplied as a separate Supplementary Information.

The Method section is now in the format requested.

REFERENCES: As a guideline, most papers should need no more than 30 references in the main text (though there is some flexibility); additional references can be cited in (and listed after) the Methods section, as detailed above. The two lists should be provided separately -- the main references after the main text, and the Methods references after the Methods.

The references are currently all listed after the Methods. We have not found a way to make this possible as some references in the main text are also used in the Method section. We will closely work with the publication office to address this problem.

MAIN TEXT STATEMENTS: We require authors to provide a detailed Author Contribution statement immediately after the acknowledgements; the specific contributions of each author must be listed. It is also a condition of publication that authors include an Author Information statement indicating how to access information regarding reprints and permissions, stating whether or not there is a financial or non-financial competing interest, and naming the author to whom correspondence and requests for materials should be addressed. **Please ensure that this section is included in the manuscript file after the Methods** (but before the Extended Data legends) - it will not appear in the print version but will appear online in the full-text HTML and PDF versions. For details of "end note" style and an example see <https://www.nature.com/nature/for-authors/formatting-guide>.

This is now updated

DATA AVAILABILITY STATEMENT: All published manuscripts reporting original research in Nature Portfolio journals must include a data availability statement. The data availability statement must make the conditions of access to the "minimum dataset" that are necessary to interpret, verify and extend the research in the article, transparent to readers. This minimum dataset may be provided through deposition in public community/discipline-specific repositories, custom proprietary repositories for certain types of datasets, or general repositories like Figshare, Zenodo and Dryad. Providing large datasets in supplementary information is strongly discouraged and the preferred approach is to make data available in repositories. More information on Nature Portfolio's reporting standards and preparing your Data Availability Statement can be found here:

<https://www.nature.com/nature-portfolio/editorial-policies/reporting-standards#reporting-requirements>

This is now updated

For all studies using custom code or mathematical algorithm that is deemed central to the conclusions, a statement must be included under the heading "Code availability", indicating whether and how the code or algorithm can be accessed, including any restrictions to access. Code availability statements should be provided as a separate section after the data availability statement but before the References. Code should be deposited in a DOI-minting repository such as Zenodo, Gigantum or Code Ocean and cited in the reference list. Authors are encouraged to manage subsequent code versions and to use a license approved by the open source initiative. Additional details can be found here:

<https://www.nature.com/nature-research/editorial-policies/reporting-standards#availability-of-computer-code>.

This is now updated

FIGURE LEGENDS: The main figure legends should be listed sequentially after the references in the main text. Each legend should begin with a brief title for the whole figure and continue with a short description of each panel and the symbols used. Any error bars in the figures must be defined (for example, s.d., s.e.m.) and the value of n indicated; see <https://www.nature.com/nature/for-authors/formatting-guide> for further explanation. I know that 1sigma is the default in astronomy, but different fields use different defaults, so we require authors to specify what they use. (Please remove the brief legends under the figures in the main text, and remove the figures from the manuscript file.)

This is now updated. Error bars derivation and explanation is given in the Methods.

DISPLAY ITEMS: Figures should be comprehensible to readers in other or related disciplines, and assist their understanding of the paper. We encourage authors who are describing complex processes to include a schematic of the main finding as part of the Extended Data to aid readers unfamiliar with the immediate discipline. Figures should be as small and simple as is compatible with clarity. All panels of a

figure should be logically connected; each panel of a multipart figure should be sized so that the whole figure can be reduced by the same amount and reproduced on the printed page at the smallest size at which essential details are visible. For guidance, Nature's standard figure sizes are 89 mm (one column), 120 mm (one and a half columns), or exceptionally 183 mm (two columns) wide; the full depth of a Nature page is 247 mm. All panels of figures should be presented on a single page and assembled into a rectangular shape for publication; please indicate any essential alignments (parts horizontal, vertical, spacings of stereo pairs, etc.).

FIGURE FORMATTING: Lettering in all figures (labelling of axes and so on) should be in uniform, sans-serif font, in lower-case type, and large enough to permit substantial reduction for publication (minimum font size 5 pt). Separate parts of a figure are labelled a, b, etc **and referred to in that way in the legends**. Units have a single space between the number and the unit, and follow SI nomenclature or the nomenclature common to a particular field. Thousands are separated by commas (1,000). Unusual units or abbreviations are defined in the legend. Scale bars rather than magnification factors should be used.

IMAGE PRESENTATION: Authors should be aware that any image provided for publication, either in print or online (as Extended Data or Supplemental Information), may be subject to a quality control process to check for image integrity and manipulation. For a full discussion of our standards regarding how images should be prepared and presented, see www.nature.com/authors/editorial_policies/image.html.

EXTENDED DATA: Nature is now integrating the supplemental figures and tables into the final version of most papers. Extended Data do not appear in the printed version of the paper but are included online within the full-text HTML and at the end of the online PDF. Extended Data are an integral part of the paper and only data that directly contribute to the main message should be presented. All Extended Data must be referred to in the main text, figure legends and/or Methods section, and their figure legends should be listed sequentially at the end of the main text, not in the Extended Data files. Authors should assemble the Extended Data into a maximum of ten, A4 size, multi-panelled display items, submitted as individual JPEG, TIFF or EPS files. They must be provided at the same quality as figures for print, but there are important differences in their formatting. More specific instructions are provided in the Extended Data Formatting Guide (http://s3-service-broker-live-19ea8b98-4d41-4cb4-be4c-d68f4963b7dd.s3.amazonaws.com/uploads/ckeditor/attachments/7823/3h_Extended_data.pdf).

SUPPLEMENTARY INFORMATION: Supplementary Information is online-only material published with the manuscript (<https://www.nature.com/nature/for-authors/supp-info>). For most papers, there should be no need for Supplementary Information beyond that already provided as Methods and Extended Data, the aim being to avoid unnecessary fragmentation of the paper online. Exceptions to this rule include large datasets that cannot be accommodated within Extended Data; video material; and more complex "Supplemental Methods" (and any associated references) that do not readily fit within the constraints of the Methods/Extended Data formats.

Please note that after the paper has been formally accepted you can only provide amended Supplementary Information files for critical changes to the scientific content, not for style. You should clearly explain what changes have been made if you do resupply any such files.

SOURCE DATA: To further increase transparency, we encourage authors to provide, in spreadsheet form, the data underlying the graphical representations used in the figures. This is in addition to our well-established data-deposition policy for specific types of experiments and large datasets. Readers of the online manuscript will be able to access the Source Data directly from the figure legend.

Spreadsheets can only be submitted in .xls, .xlsx or .csv formats. One file per figure is permitted; thus, if there is a multi-panelled figure the Source Data for each panel should be clearly labeled in the csv/Excel file; alternatively the data for a figure can be included in multiple, clearly labeled sheets within an Excel file. File sizes of up to 30 MB are permitted; however, it is expected that the vast majority of Source Data files will be considerably smaller than this. When submitting these files with your manuscript, please select the file type “Source Data” and use the title field in the file description tab to indicate the figure to which the Source Data pertain.

In my previous report I wrote that all the data to produce the figures are included in the Tables of the manuscript. Is a Source Data file necessary? I will provide that after submission in case it is needed.

Nature is committed to improving transparency in authorship. As part of our efforts in this direction, we are now requesting that all authors identified as ‘corresponding author’ create and link their Open Researcher and Contributor Identifier (ORCID) with their account on the Manuscript Tracking System prior to acceptance. ORCID helps the scientific community achieve unambiguous attribution of all scholarly contributions. You can create and link your ORCID from the home page of the Manuscript Tracking System by clicking on ‘Modify my Springer Nature account’ and following the instructions in the link below. If you experience problems in linking your ORCID, please contact the Platform Support Helpdesk.

This information is available.

Reviewer Reports on the Second Revision:

Referees' comments:

Referee #1 (Remarks to the Author):

I have read the revised version of the manuscript and also the companion paper which details the photometric redshift measurement and constraints. With that, my previous comments and concerns have been met and I am happy to recommend the paper for publication in Nature.

Referee #2 (Remarks to the Author):

I have looked through the revised manuscript, and I am happy with the revisions. I recommend this article for publication.